# Spatial heterogeneity ensures long-term stability in vegetation and *Fritillaria meleagris* flowering in Uppsala Kungsäng, a semi-natural meadow

Håkan Hytteborn[1], Bengt Å. Carlsson[1], Brita M. Svensson[1], Liquan Zhang[2], Håkan Rydin[1]*

1 Department of Ecology and Genetics, Evolutionary Biology Centre, Uppsala University, Uppsala, Sweden,
2 State Key Laboratory of Estuarine and Coastal Research, East China Normal University, Shanghai, China

* hakan.rydin@ebc.uu.se

**Data Availability Statement:** Data are available in S2 and S5 Appendices. Weather data for Ultuna weather station are available at https://www.ffe.slu.

## Abstract

Semi-natural grasslands are becoming increasingly rare, and their vegetation may be affected by environmental changes and altered management. At Kungsängen Nature Reserve, a wet to mesic semi-natural meadow near Uppsala, Sweden, we analysed long-term changes in the vegetation using data from 1940, 1982, 1995 and 2016. We also analysed the spatial and temporal dynamics in the *Fritillaria meleagris* population based on countings of flowering individuals in 1938, 1981–1988 and 2016–2021. Between 1940 and 1982 the wet part of the meadow became wetter, which led to an increased cover of *Carex acuta* and pushed the main area of flowering of *F. meleagris* up towards the mesic part. Annual variation in the flowering propensity of *F. meleagris* (in May) was affected by temperature and precipitation in the phenological phases of growth and bud initiation (June in the previous year), shoot development (September in the previous year) and initiation of flowering (March–April). However, the response to weather was in opposite directions in the wet and mesic parts of the meadow, and the flowering population showed large year-to-year variation but no long-term trend. Variation in management (poorly documented) led to changes in different parts of the meadow, but the overall composition of the vegetation, species richness and diversity changed little after 1982. Species richness and species composition of the meadow vegetation, and the long-term stability of the *F. meleagris* population are maintained by the variation in wetness, highlighting the importance of spatial heterogeneity as an insurance against biodiversity loss in semi-natural grasslands and nature reserves generally.

## Introduction

Traditionally in large parts of Europe, managed hay meadows (used for winter fodder production and aftermath grazing) formed an essential foundation in the agricultural system up until the agricultural revolution [1–4]. Meadows along river banks and sea shores are an exception

se/lm. Species indicator values are available in: Tyler T, Herbertsson L, Olofsson J, Olsson PA. Ecological indicator and trait values for Swedish vascular plants. Ecol Indic 2021;120: 106923. doi: 10.1016/j.ecolind.2020.106923.

**Funding:** The authors received no specific funding for this work.

**Competing interests:** The authors have declared that no competing interests exist.

to the rule that semi-natural grasslands generally are nutrient-poor since the yearly flooding brings new nutrients to the meadow [1]. However, small elevational differences lead to spatial variation in flooding, and thereby also to spatial heterogeneity in nutrient availability, soil aeration and related factors.

Thanks to their nutrient-rich soil, floodplain meadows have been important for sustainable production by a management that usually involves mowing in mid to late summer followed by aftermath grazing [5]. With the introduction of commercial fertilisers, the vast acreage of meadows was no longer needed [4], and they were converted to pastures, arable fields, forested or simply abandoned (with overgrowth as the result). Apart from remaining meadows being few and far between, management practices have weakened, resulting in a general loss of species diversity and a more trivial flora [6]. In addition, anthropogenic nitrogen deposition has led to further decreases in plant species diversity, as this favours competitive and other nutrient-demanding species, which also leads to increasing litter production [7,8]. By shading and preventing seeds from reaching the soil surface, this increase in biomass and litter makes it more difficult for plants to establish [9]. However, some meadows still retain high conservation values, regarding both their cultural heritage and biodiversity values, not only for plants but also for fungi, insects and birds [10; www.nationalredlist.org]. Semi-natural grasslands are also important for the delivery of ecosystem services such as water regulation, pollination and carbon storage [11].

Prior to the last decades of the nineteenth century there were vast areas of flooded meadows on the plains along the River Fyrisån, both north and south of the city of Uppsala, south-central Sweden [12]. The only remnant today is Uppsala Kungsäng, hereafter referred to by its locally used name 'Kungsängen'–'the King's meadow', established as a nature reserve in 1951. In this paper we use unique long-term data to discuss the dynamics in the vegetation of Kungsängen over eight decades.

The 12.5-ha reserve is situated about 3 km south of Uppsala, surrounded by fertilised pastures and arable fields. Elevation varies between 1.2 and 2.2 m a.s.l. in the studied area which creates a spatial heterogeneity in wetness (Fig 1). The soil is an alluvial clay [12], partly a gyttja clay [13]. The meadow's groundwater generally follows the water level in the river [12].

The history of hydrology and management is complex (see S1 Appendix for details). After land uplift, vast meadow systems were created by iron-age farmers, remaining thereafter under continuous use [14: p. 206]. Kungsängen itself was below sea level prior to ca. AD 700 [15], and increasing parts became available for fodder (mainly *Equisetum fluviatile* and *Carex* spp.) from around 1600 [15], although seashore plants such as *Glaux maritima* and *Triglochin maritima* were still present [16] (nomenclature follows Euro+Med Plantbase [17]). The meadow flooded each spring, and also, in some years, during autumn. More recently the water level in the river increased slightly between 1940 and 1970 [18].

Traditionally, mowing was carried out in late July (later in wet years) followed by aftermath grazing [15]. In 1942, Sernander [19: p. 23] observed that grazing took place throughout the whole summer. As Kungsängen was set aside as a nature reserve in 1951 [20], it was stipulated that the meadow should be mown annually with aftermath grazing.

However, subsequent management has varied, and is poorly documented, but in 1979–1983 mowing occurred around 24 June [21]. The wettest parts were only mown in some years, allowing the sedges to form tall tussocks that had to be cut by rotor cultivator in the 1990s (S.-O. Borgegård, unpublished manuscript). Following a new management plan in 1999 [22], the meadow is mown in early August, followed by cattle grazing.

The vegetation of Kungsängen has been investigated several times: the first in 1938–1940 [14: pp. 147–165], repeated and extended in 1982 [21]. We now add results from 1995 and 2016, allowing an analysis of vegetation changes over eight decades. Changes in vegetation

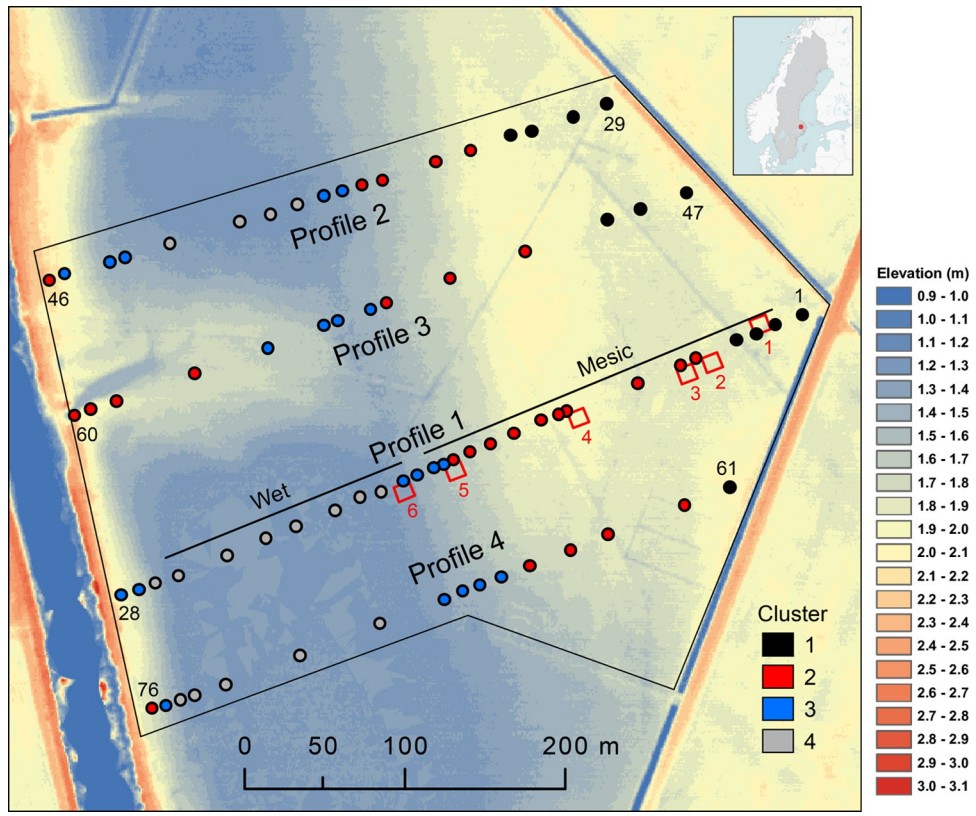

**Fig 1. Kungsängen nature reserve with the River Fyrisån to the left.** Vegetation plots (2 m × 2 m) were placed along four profiles (Profile 1 followed since 1940, Profiles 2–4 since 1982), colours identify the four vegetation clusters identified in 1982. For profile 1, the Wet and Mesic parts identified in 1940 are indicated. Flowering *Fritillaria meleagris* were counted in six plots, 1–6, (red squares, 10 m × 10 m). Background colours show elevation based on laser scanning with 1 m grid (Markhöjdmodell grid 1+) from Lantmäteriet (the Swedish Land Survey, www.lantmateriet.se). Plot elevation varies between 1.2 and 2.2 m a.s.l. Inset shows location in Sweden. The area treated by rotor cultivator in the 1990s roughly corresponds to Cluster 4.

since the 1940s could be hypothesised to concern species with specific preferences regarding grazing and mowing. For example, low management intensity might favour large competitive species and be disadvantageous for rosette-forming, light-demanding and early-flowering species. Combined changes in management and water level may in turn also affect the availability of mineral nutrients, and in concert with airborne nitrogen deposition (which increased until ca 1990 and then decreased; [23]), lead to changes in species composition.

One notable plant species in the meadow is the bulbous geophyte *Fritillaria meleagris* (Liliaceae). The species is native to central and eastern Europe, with scattered localities further east to Altai [24]. In central Europe, the species is considered to be vulnerable (VU; [25]). Sten Carl Bjelke found *F. meleagris* at Kungsängen in 1742 or 1743 where Linnaeus noted it in 1755 as 'copiosissime' [19: p. 94]. It was thought to have been dispersed with wastes from nearby Uppsala, perhaps from the botanical garden [26]. Today it is probably the largest population in Sweden (Sebastian Sundberg, pers. comm.), reflected in its Swedish name 'kungsängslilja'. The spectacular flowering of *F. meleagris* in May attracts thousands of Uppsala citizens, and the species is used in the logotype of Region Uppsala. It appears with purple (most common) and white flowers (and with a few pink intermediates). The white morph was preferred by flower-picking townsfolk [19], which may have decreased its relative frequency in the population by reducing recruitment from seeds.

Earlier studies have suggested that yearly weather variations can cause variation in flowering propensity in long-lived perennials (e.g. [27]). For *F. meleagris*, several authors have shown that variation in flowering is related to variation in wetness [18,28–30]. We will here present long-term trends and annual variation in flowering in *F. meleagris* with data going back to 1938, and relate this to weather variation and spatial heterogeneity in wetness on Kungsängen.

*Fritillaria meleagris* seeds are water-dispersed and require a cold period to germinate [21]. It probably takes several years from seed to mature bulb. In early spring, the bulb sprouts a stalk along with scale-like, ground-level leaves, and above those up to eight normal leaves from the stalk [21]. Flowers emerge in May and are cross-fertilised [31], but according to Zych et al. [32], selfing (rarely occurring in natural populations) results in fully developed seeds. Seeds are enclosed in a capsule which matures in June and July after which the stalk withers [18]. In years with early mowing the seeds will end up in the hay, rather than being dispersed [21].

After senescence in June, the previous year's bulb is replaced by a new bulb formed by the two inner low leaves [21]. After summer dormancy, a shoot bud forms that develops a stalk in late August, which reaches the soil surface in early November. The stalk, unexpanded leaves and a pre-formed flower bud then enter winter dormancy [21,33]. The species thus has three periods that we assume to be particularly influenced by environmental conditions: June in the previous year (growth and bud initiation), September (shoot development) and March–April (initiation of flowering).

In perennial plants, the allocation of resources to reproduction may diminish future chances to flower. Such a cost of reproduction is difficult to demonstrate in natural populations, as it is often confounded by year-to-year variation in resources and weather conditions [34]. The probability of flowering is often related to plant size (e.g. [35]). Accordingly, Zhang [21] found that flowering *F. meleagris* at Kungsängen originated from a larger bulb in the spring than vegetative individuals, while in summer they produced a smaller new bulb for next year. Thus, one could hypothesise that many flowering individuals will be vegetative in the following year, or even enter prolonged dormancy (i.e. skip one or several growing seasons; [36]), a feature shared with many other bulbous plants [36].

## Aims

We will view the vegetation composition found in the 1940 survey as a reflection of traditional management of a wet–mesic meadow. Since then, the management has periodically adopted the modern practice of early hay harvest, as well as being mechanised, and we hypothesise that this (in combination with other environmental changes, such as nitrogen deposition) may have led to losses in species richness and plant diversity and to changes in community composition.

For *F. meleagris* we ask whether population size (reflected by flowering) shows a trend over the last eight decades, or whether variation in flowering can be explained by between-year weather fluctuations, and whether flowering in one year is affected by flowering in the preceding year. To be able to discuss the importance of flowering and other life stages for the population dynamics, we performed a sensitivity analysis based on demographic data from Zhang [21]. Finally, we tested whether the relative frequency of the white morph has increased after 1940, as picking flowers at Kungsängen is now prohibited.

## Materials and methods

### Field methods

In June 1940, Gustaf Sandberg recorded the vegetation in twenty-eight 2 m × 2 m plots along a profile across Kungsängen (Profile 1, Fig 1, details in Sandberg [14: pp. 149–150]). Detailed

**Table 1. Total species richness (S) and inverse Simpson diversity (D) in Profile 1 (1940–2016) and all profiles (1982–2016).** Note that in 1940 only Profile 1 was surveyed.

| Survey year | Source | Profile 1 (Plots 1–28) | | Profiles 1–4 (Plots 1–76) | |
| --- | --- | --- | --- | --- | --- |
| | | **S** | **D** | **S** | **D** |
| 1940 | Sandberg [17] | 63 | 11.8 | Not surveyed | |
| 1982 | Zhang [18] | 72 | 16.3 | 78 | 20.4 |
| 1995 | This study | 61 | 17.3 | 69 | 21.1 |
| 2016 | This study | 73 | 16.0 | 91 | 21.2 |

descriptions, with landmarks and distances along the profile, made it possible to later relocate these plots within a few metres. In 1982, Zhang [21] recorded the vegetation in these plots, and in three additional parallel profiles (Profiles 2–4, plots 29–77; Fig 1). We repeated these analyses in 1995 and 2016. Hence, one dataset is composed of the 28 plots in Profile 1, surveyed in 1940–2016, and a second dataset consists of all 76 plots in Profiles 1–4, surveyed in 1982–2016 (Table 1).

Following Sandberg [14], all surveys recorded species cover using the 5-degree Hult-Sernander-Du Rietz scale, where a score of 5 indicates a cover of 50% or more, 4 a cover less than 50% but over 25%, etc., and 1 a cover of 6.25% or less (data in S2 Appendix).

In 1938, Sandberg [19: p. 98] counted the number of flowering *F. meleagris* in six 10 m × 10 m plots along Profile 1 (Fig 1). We repeated the census in 1981–1988 and in 2016–2021. The individuals were classified according to their colour as purple, pink or white. The 2016 census began too late to distinguish the different colour morphs.

## Statistical analyses

**Changes in vegetation.** Plot elevation was extracted from laser scanning data with 1 m grid (Markhöjdmodell grid 1+) from Lantmäteriet (the Swedish Land Survey, www.lantmateriet.se), using ArcGis v. 10.8 [37].

Vegetation data were analysed using the 'vegan' package [38] in R v. 4.2.1 [39]. The analyses included ordination by non-metric multidimensional scaling (NMDS). To define vegetation clusters we calculated a dissimilarity matrix with 'vegdist' (using Bray-Curtis as dissimilarity measure), then created a hierarchical cluster dendrogram with 'hclust' (with 'ward.D2' as the agglomeration method), and 'cutree' to cut the dendrogram into clusters. In ordination diagrams we enclosed groups of plots by convex envelopes with the 'ordihull' function, and used 'ordisurf' to overlay a Generalised Additive Model (GAM) of elevation on the ordination graph. Diversity was calculated as the inverse Simpson index ($D = 1/\sum p_i^2$).

Differences in species cover were tested with an Anova with Survey year as predictor and Plot as random factor using the 'lme4' package in R [40], with packages 'pbkrtest' [41] and 'lmerTest' [42] for posterior tests of differences among surveys. The class midpoints in the Hult-Sernander-Du Rietz scale were used to transform the 5-degree scale data into percent cover. We tested differences in plot frequency for the species between the first and last survey (1940 vs 2016 for Profile 1; 1982 vs 2016 for all profiles). We used a binomial test of the null hypothesis that the number of plots a species had occupied was equal to the number of plots from which a species had disappeared.

As proxies for ecological conditions we used ecological indicator values for Swedish vascular plants [43]. We calculated changes in the species indicator values using abundance-

weighted value for plot $j$ following Diekmann [44],

$$Weighted\ average = \sum_{i=1}^{n}(r_{ij} \times x_i)/\sum_{i=1}^{n} r_{ij}$$

where $r_{ij}$ is the cover of species $i$ (abundance scale 1–5) in sample plot $j$, and $x_i$ the indicator value of species $i$.

Six indicators were thought to help explain changes in species composition over time [43]. These indicator values express the degree of positive reaction to Moisture (scale 1–12), Light (1–7), Nitrogen (1–9), Grazing/mowing (1–8, where 1, does not endure grazing and mowing; 4, neutral; 8, demands repeated or continuous grazing or mowing), Soil disturbance (1–9, where 9 indicates required yearly disturbance) and Soil reaction (1–8, where 1 is a pH < 4.5 and 8 is pH > 7.5). The Phosphorus indicator was not used, as it was correlated with the Nitrogen indicator ($r = 0.72$). We used 'envfit' within the vegan package to fit indicators as environmental vectors onto the NMDS ordinations. Differences in indicator values between groups or clusters identified in the ordination were tested with mixed repeated-measures Anova with Survey year as a repeated effect, Plot identity as a random effect and Cluster as a fixed effect (using SPSS v. 28 [45]).

**Variation in flowering in *Fritillaria meleagris*.** To relate the number of flowers to weather we used monthly values for temperature and precipitation in the three active periods– June and September in the previous year, and March–April (just before flowering) in the current year–from the Ultuna weather station, ca 2 km south of Kungsängen (https://www.ffe.slu.se/lm). Temperature and precipitation were uncorrelated ($p = 0.35$, 0.66 and 0.99 for the three periods, respectively). We used a generalised linear mixed model (GLMM) to evaluate the interactive effects of climatic variation and topography on the number of flowers produced per plot in a given year. Because our response variable (number of flowers) was discrete count data, we used a model with Poisson errors and we included a random effect for plot to account for plot-specific intercepts. Specifically, our model took the form:

$$Log(N_{py}) \sim clim_y + elev_p + clim_y \times elev_p + \theta_p + \varepsilon$$

where $N$ is the number of flowers produced in plot $p$ during year $y$, $clim$ is the climate for year $y$, $elev$ is the elevation of plot $p$, $\theta$ is a random intercept for plot $p$, and $\varepsilon$ is the Poisson error term. We centred and scaled predictor variables prior to analysis by subtracting their mean and dividing by their standard deviation [46]. We used the 'glmer' function of the lme4 package [40] to fit the model in R. We computed the marginal and conditional $R^2$ for GLMMs using the 'MuMIn' package [47].

A similar model was used to test the effect of the number of flowers in the previous year (log-scaled), again in combination with elevation and with plot as a random factor. Since data for the previous year was lacking for 1938, 1981 and 2016 we eliminated these years, so that this test was based on 12 years' data.

To be able to discuss the importance of the various life stages on population growth rate, we performed a sensitivity analysis based on demographic data from Zhang [21]. We constructed transition matrices (S6 Appendix) from which we calculated long-term population growth rates and sensitivity values for the yearly transitions 1981 to 1982 and 1982 to 1983. We used the following five stages: small, medium, large vegetative, reproductive and dormant. We included a dormant stage, since prolonged dormancy was observed in the field by Zhang ([21]; cf. [36]). Including such a cryptic stage will give more accurate results [48]. Into this stage we placed plants that were present in 1981 and 1983, but not seen in 1982. We assumed that similar proportions of the population are in the dormant stage each year. We also assumed that

newly discovered plants originated from plants in the dormant stage irrespective of which stage they entered. Plants found in 1981 but never relocated were assumed dead. Analyses were made using MATLAB [49].

## Ethics statement

We obtained permission from the Uppsala County Board to work in the Kungsängen Nature Reserve.

## Results

### Changes in vegetation

At Kungsängen today we find species typical of wet–mesic mown and grazed meadows. The sea-shore plants still growing at Kungsängen at the time of Linnaeus have all disappeared except *Alopecurus arundinaceus* and its hybrid with *Alopecurus pratensis*. There are no rare or red-listed plant species found on the meadow today.

**Profile 1, 1940–2016.** In Profile 1 the total number of species varied over time between 61 and 73 and was somewhat lower in 1940 and 1995 than in 1982 and 2016 (Table 1). There were 8 species with evidence for a net decrease between 1940 and 2016, and 14 with an increase (S7 Appendix). In 1940 there was a strong dominance of the grasses *Deschampsia cespitosa* (22% of the total abundance), *Poa pratensis*, *Elytrigia repens* and *A. pratensis*, but from 1982 the obvious dominant was *Carex acuta* (from 6% in 1940 to 16% in 1982–2016).

In the ordination of Profile 1 we distinguished plots that were separated according to elevation into a Mesic group in 1982 (plots 2–16, elevation ≥ 1.55 m a.s.l., Fig 1) and a Wet group (18–25, elevation ≤ 1.35 m a.s.l.). Five plots were not assigned to a group (plot 1 is somewhat disturbed by trampling, plots 26–28 were on the elevated levee by the river and plot 17 fell between the distinct groups).

In the species ordination (Fig 2B), aquatic plants, such as *Alisma plantago-aquatica*, *C. acuta*, *E. fluviatile*, *Lemna* spp. and *Utricularia vulgaris*, are found on the right. Slightly further to the left we find plants characteristic of wet meadows, such as *Stellaria palustris* and *Thalictrum flavum*. In the lower part *Carex cespitosa* and *Filipendula ulmaria* are found. *Fritillaria meleagris* has a central position together with the majority of other meadow herbs and grasses, such as *A. pratensis*, *Avenula pubescens*, *Lathyrus pratensis* and *Ranunculus acris*. In the upper left part several ruderal species are placed, such as *Capsella bursa-pastoris* and *Plantago major*.

The indicator values were (irrespective of survey) significantly higher for plots in the Wet group than in the Mesic group regarding Moisture and Light (Fig 3; S3 Appendix [analyses with p-values]; and Fig 2A [vectors in inset graph]), and lower regarding Grazing/mowing indicator values.

The vegetation in the **Wet group** changed considerably between 1940 and 1982, The most conspicuous change was the strong increase in cover of *C. acuta* between 1940 and 1982 (Fig 4A). This led to a more clear separation of the two groups along NMDS1 (Fig 2A), and was concomitant with an increase in the moisture indicator value (Fig 3). Nearly half of the species found in 1940 were not found in the three following surveys, e.g. *D. cespitosa*, *E. repens* and *P. pratensis*, and the number of species decreased from 28 in 1940 to 17 in 2016. Throughout the entire study period, the group was characterised by wet meadow species such as *A. arundinaceus*, *Caltha palustris*, *Carex disticha* and *Galium palustre*, and from 1982 also *C. cespitosa*, *E. fluviatile*, *F. ulmaria* and *Persicaria amphibia*.

The **Mesic group** was dominated by grasses (*A. pratensis*, *D. cespitosa*, *Festuca rubra*, *Phleum pratense*, *P. pratensis* and *Schedonorus pratensis*, and from 1982 *A. pubescens*). Among the many herbs, *L. pratensis*, *Ranunculus auricomus*, *R. acris*, *Rumex acetosa*, *Trifolium repens*

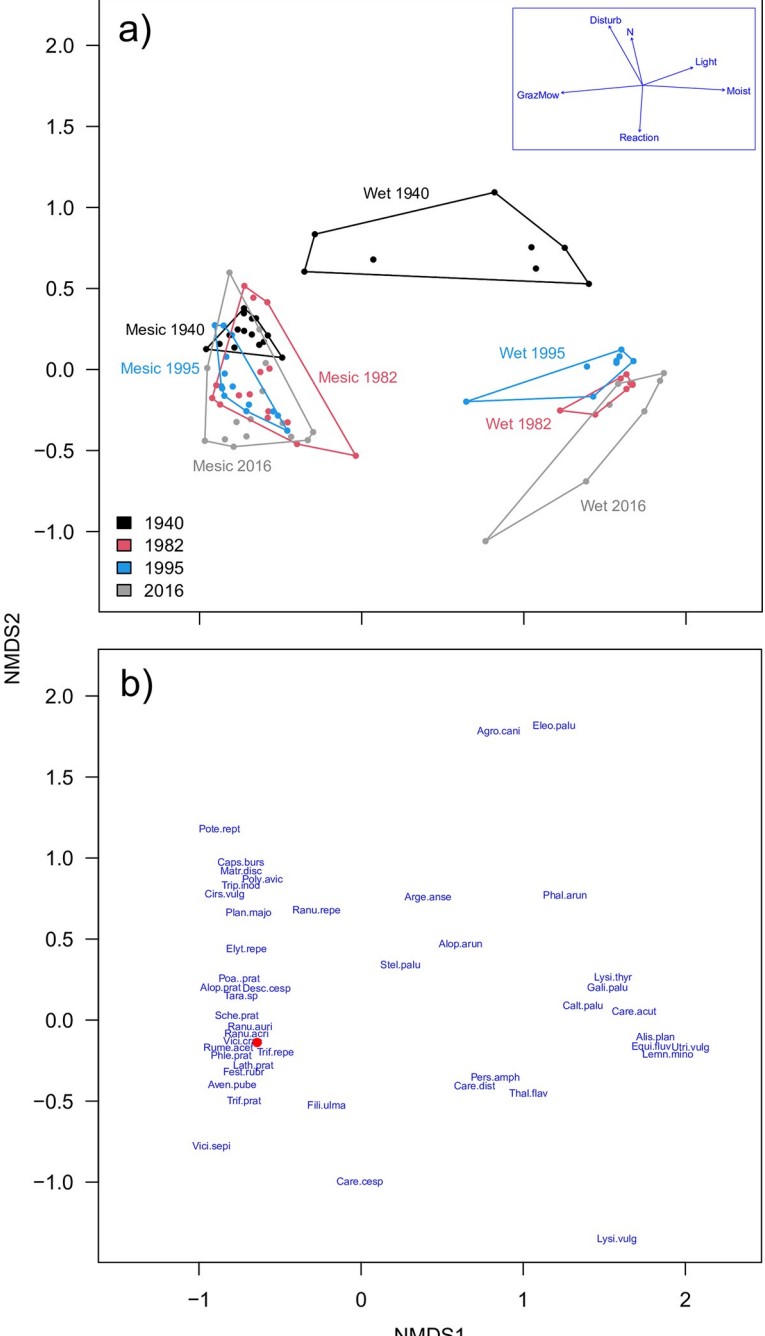

**Fig 2.** (a) NMDS ordination of plots in Profiles 1 on Kungsängen in 1940, 1982, 1995 and 2016. mesic plots, *n* = 15; wet plots, *n* = 8). NMDS stress = 0.07. Inset shows environmental vectors based on abundance-weighted indicator values for each plot. (b) Species ordination, showing the six most abundant species in each envelope and other representative species that are mentioned in the text (such as ruderals). The red dot is *Fritillaria meleagris*. Some species' positions are slightly shifted to reduce text overlap. For full species names, see S2 Appendix.

and *Vicia cracca* were found in all four surveys in relatively high frequency. The most conspicuous change was the gradual decrease of *D. cespitosa*. This led to a more equal distribution of cover among species (Fig 4A) and more variable vegetation (increasing envelopes in Fig 2A).

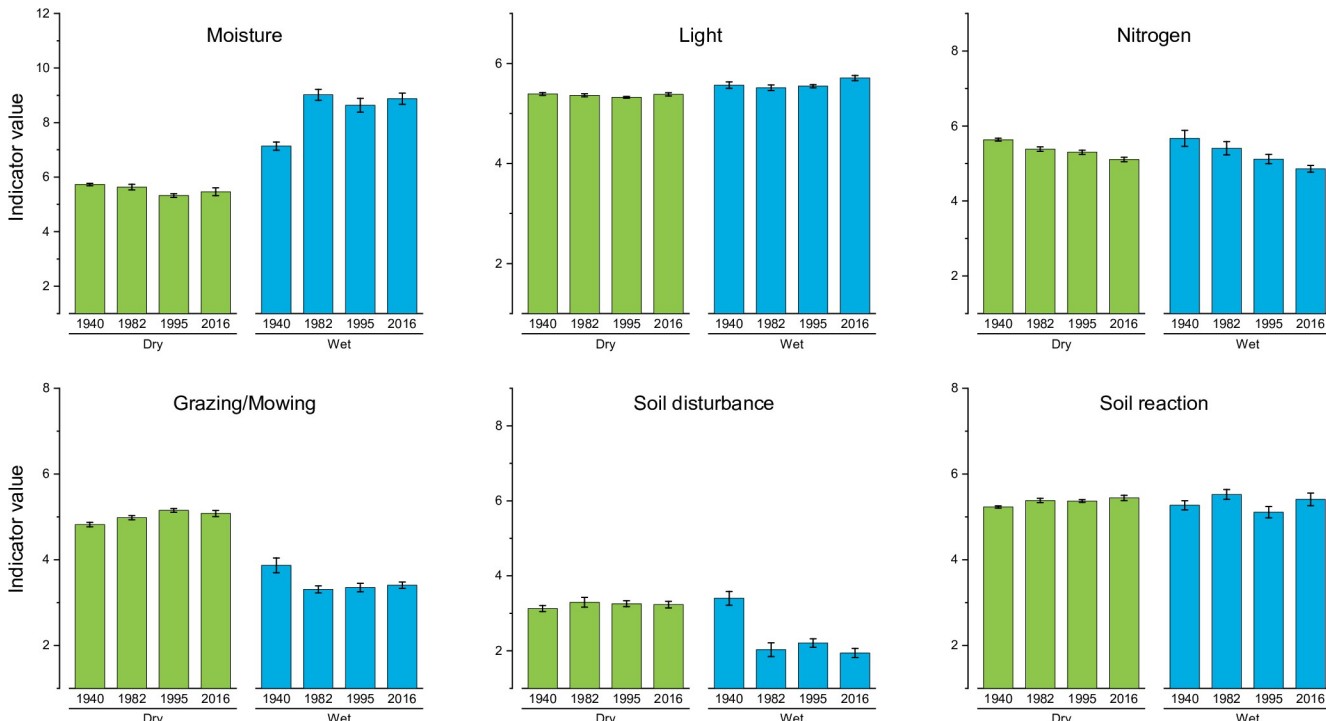

**Fig 3. Estimated marginal means ± 1 S.E. for abundance-weighted indicator values for plots in Profile 1 regarding Moisture, Light, Nitrogen, Grazing/mowing, Soil disturbance and Soil reaction in 1940, 1982, 1995, and 2016.** *N* for the two groups are: Mesic, 15; Wet, 8. The range of each *y*-axis covers the full range of the respective indicator. Results from the mixed repeated-measures Anova are presented in S3 Appendix.

*Fritillaria meleagris* occurred mainly in this group, and did not change much in cover over time (mean cover in the 5-degree scale was 0.73, 0.87, 0.87 and 0.73 for the four surveys, respectively).

Both groups moved downwards along NMDS2 (especially the Wet group; Fig 2A). This axis was positively associated with Soil disturbance and Nitrogen, and negatively with Soil reaction (Fig 2A, inset). This is consistent with the decrease in Soil disturbance indicator in the Wet group between 1940 and 1982, and in the continuous decrease in the Nitrogen indicator from 1940 to 2016 in both groups: species with high nitrogen indicator values (e.g. *D. cespitosa*, *E. repens*, *P. pratensis*, *Ranunculus repens*, *R. auricomus*) have been partly replaced by those with slightly lower values (*C. acuta*, *C. cespitosa*, *P. pratense*, *Stellaria graminea*).

**Profiles 1–4, 1982–2016.** When all four profiles are considered for 1982–2016, species richness was still lowest in 1995 (69) and highest in 2016 (91) but diversity (D) did not vary much (Table 1). There were 33 species with a net decrease between 1982 and 2016 (but only 6 of these had cover >1% in any survey), and 14 with an increase (S7 Appendix). The two species with the highest cover over the whole study were *A. pratensis* (13%) and *C. acuta* (12%). Among the other eight species with highest cover in 1982, *C. disticha*, *D. cespitosa*, *F. rubra* and *P. pratensis*, were also top ten in 1995, as were *C. cespitosa*, *C. disticha* and *F. rubra* in 2016 (Fig 4B).

To be able to follow changes in different parts of the meadow we first performed a cluster analysis of the starting point, i.e. the 1982 recordings. We identified four clusters that were (i) spatially separated in the meadow (Fig 1), (ii) well separated in the ordination as described below (Fig 5A) and (iii) reflect the elevation gradient (mean elevation for Clusters 1–4 was 2.0, 1.9, 1.5 and 1.3 m a.s.l., respectively).

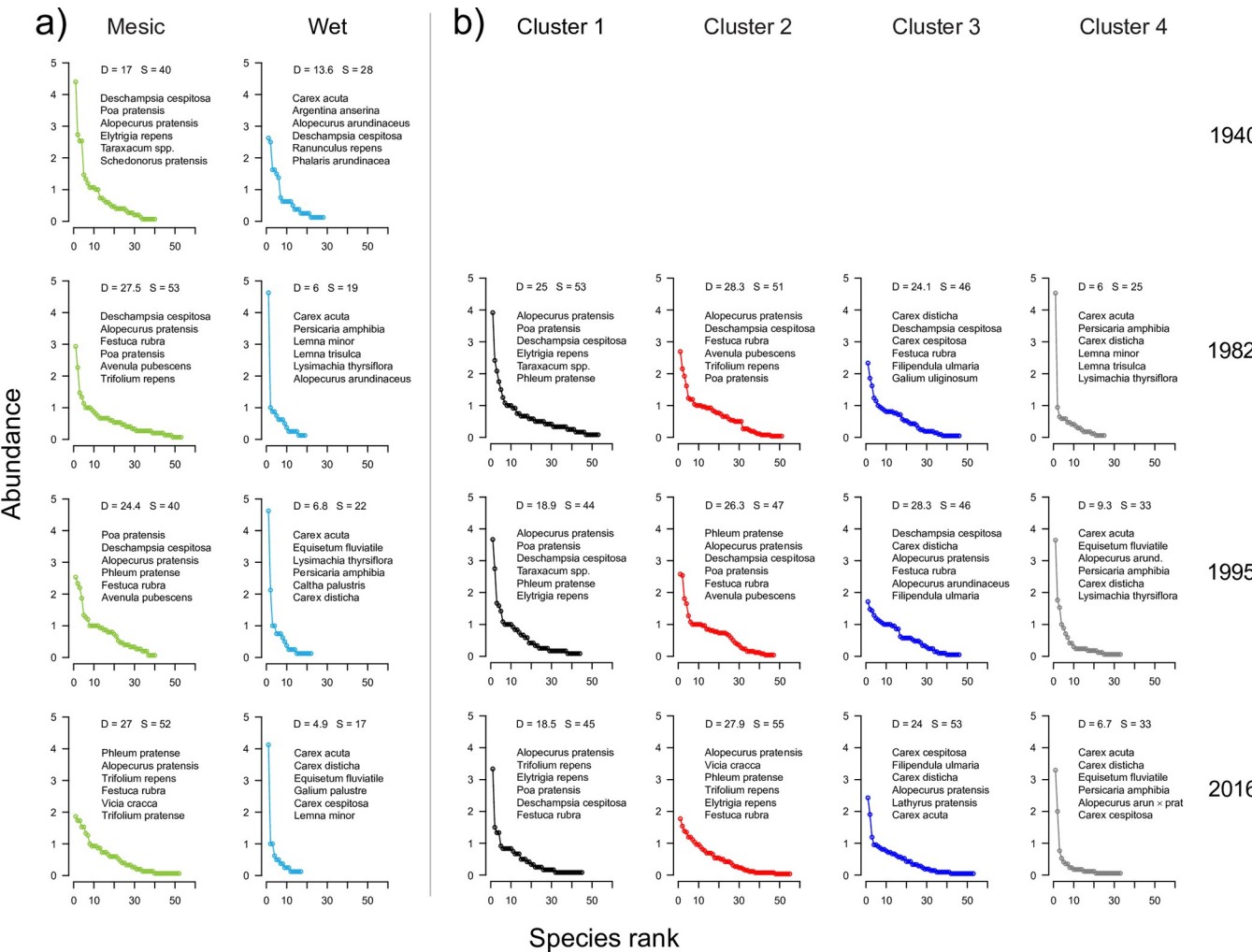

**Fig 4. Rank-abundance curves showing each species' mean cover (scale 1–5), with the most abundant species listed in rank order.** D, Simpson inverse index; S, species richness. (a) Profile 1 (1940–2016) divided into the Wet and Mesic parts. (b) Profiles 1–4 (1982–2016) divided into Clusters 1–4. For full species names, see S2 Appendix.

In the ordination of all surveys we follow the changes within each starting cluster. In the species plot (Fig 5B) we find wetland plants such as *C. palustris*, *Carex vesicaria*, *C. acuta* and *E. fluviatile* to the right. Ruderal plants such as *C. bursa-pastoris* and *Poa annua* are found towards the upper left, and in the centre are generalist meadow plants such as *A. pratensis*, *A. pubescens*, *D. cespitosa*, *P. pratensis*, and also *F. meleagris* (Fig 5B). Clusters 1, 2 and 4 moved towards the left in the biplot after 1982 (Fig 5A), while changes along axis 2 were small. Over the years the clusters have become more diverse (larger envelopes) and less well separated (larger overlap among envelopes).

**Cluster 1** (12 plots; 53, 44 and 45 species in the surveys 1982, 1995 and 2016). The plots in this cluster are situated near the entrance or along the borders of the meadow (Fig 1). In addition to the dominant meadow generalists the cluster is characterised by ruderal or weedy species such as *C. bursa-pastoris*, *E. repens*, *Matricaria discoidea*, *P. major*, *Polygonum aviculare*, *Taraxacum spp.*, *Tripleurospermum inodorum* and *Urtica dioica*. From 1982 to 1995 there was an increased cover of *F. rubra*, *L. pratensis* and *T. repens* and a decrease in

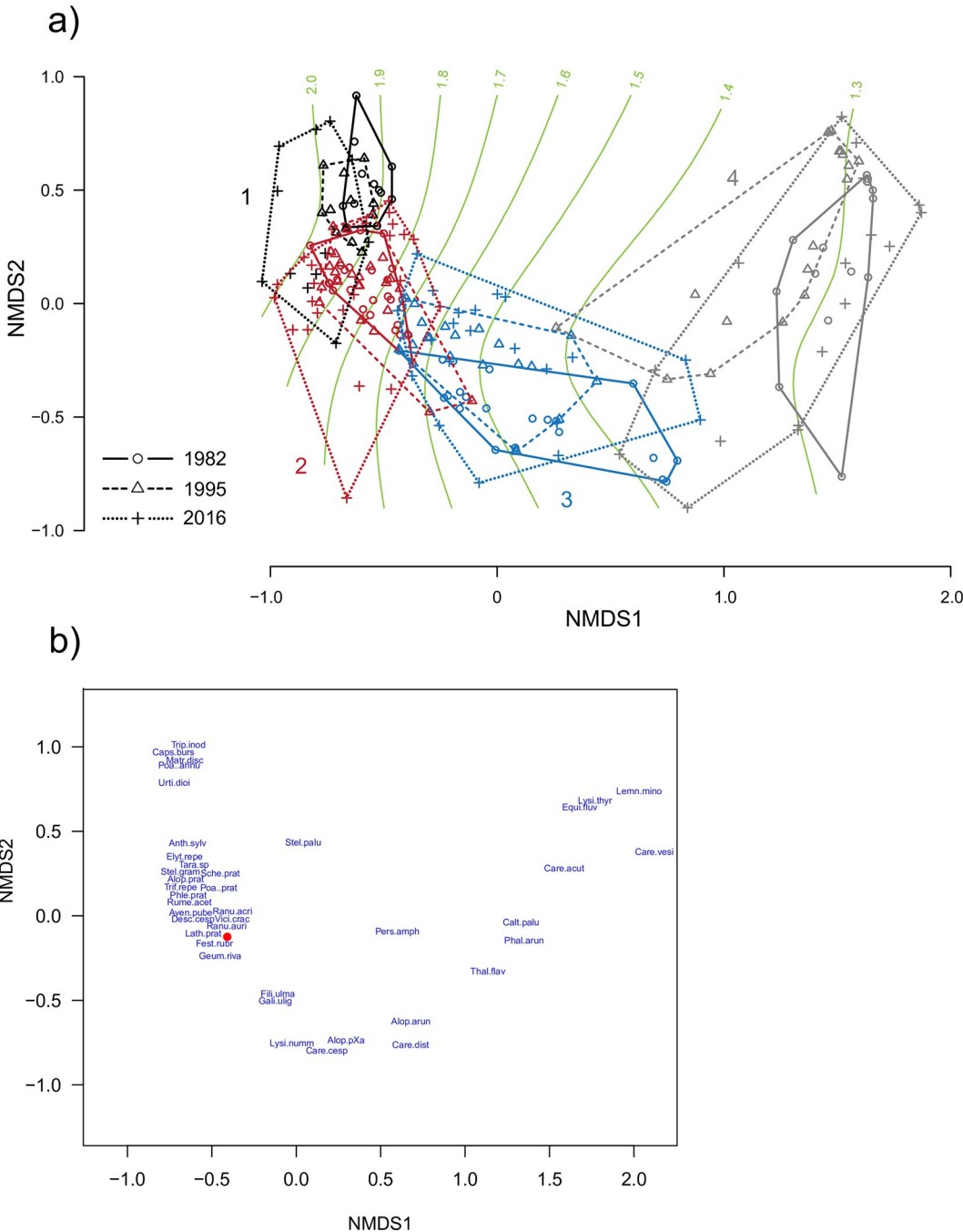

**Fig 5.** (a) NMDS ordination of plots in Profiles 1–4 on Kungsängen in 1982, 1995 and 2016. The plots are allocated among four clusters based on the vegetation in 1982 (*n* = 76). These clusters are identified by colours. The three survey years are identified by symbols and line types. NMDS stress = 0.07. The green lines are fitted elevation contours (m a.s.l.). (b) Species ordination, showing the six most abundant species in each envelope and other representative species that are mentioned in the text (such as ruderals). The red dot is *Fritillaria meleagris*. Some species positions are slightly shifted to reduce text overlap. For full species names, see S2 Appendix.

ruderals, and in 2016 the cluster had gradually become more similar to Cluster 2 (larger overlap in Fig 5A).

**Cluster 2** (26 plots; 51, 47 and 54 species in three surveys). Besides grasses such as *D. cespitosa* (decreasing over time; Fig 4B), *A. pratensis* and *A. pubescens*, this cluster was characterised by many herbs, e.g. *Geum rivale*, *L. pratensis*, *R. acris*, *T. repens* and *V. cracca*. The increased similarity with Cluster 1 is exemplified by the increase in *E. repens*. In 1982, the cover of *F. meleagris* was highest in this cluster, but in 1995 it was similar, and in 2016 somewhat higher, in Cluster 3.

**Cluster 3** (21 plots; 46, 46 and 53 species in the three surveys) represents a transition between the mesic and wet parts of the meadow (Figs 1 and 5A). Several species indicating moist conditions are shared with Cluster 4 (e.g. *C. disticha* and *F. ulmaria*; Fig 4B), whereas *A. pratensis* is shared with Clusters 1–2. A characteristic species was *C. cespitosa*, but it had lower cover in 1995 than in 1982 or 2016. As in Clusters 1 and 2, *D. cespitosa* was important, but had decreased considerably in 2016.

**Cluster 4** (17 plots; 25, 33 and 33 species in the three surveys) represents the wet, species-poor part, dominated by *C. acuta*. In 1982 it was well separated (Fig 5A), but later showed some overlap with Cluster 3 as the dominance of *C. acuta* decreased (Fig 4B), and species richness increased. *C. disticha* and *P. amphibia* were common during the whole study, whereas a number of wet meadow and aquatic species (e.g. *E. fluviatile*, *Lemna* spp. and *Lysimachia thyrsiflora*) varied in importance over time.

The largest difference in indicator values among clusters was for Moisture, which ranged from just over 5 in Clusters 1 and 2 to almost 9 in Cluster 4 (Fig 6, S4 Appendix). The opposite order was found for Soil disturbance and Grazing/mowing, where Clusters 1 and 2 had the highest values and Cluster 4 the lowest. Changes over time were small and were found only for Moisture (a slight decrease over the years, particularly in Cluster 4) and Nitrogen (a notable decrease in Cluster 4 between 1995 and 2016; Fig 6, S4 Appendix).

## Variation in flowering in *Fritillaria meleagris*

*Fritillaria meleagris* is most abundant in the mesic parts of Kungsängen (Figs 2B and 5B). The total number of flowering plants in the six plots (each 10 m × 10 m) varied between 1695 (1982) and 4519 (2021), with an average over the 15 censuses (from 1938 to 2021) of 2683 (Fig 7, S5 Appendix). The variation was considerable between consecutive years, but without any increasing or decreasing trend over time (Fig 7). The number of flowering individuals in 1938 was highest in plot 5, but in the 1980s highest in plot 4 (Fig 8). From 2016, plot 5 again was highest in most years.

The proportion of white flowers varied between 2.4 and 4.5% (mean = 3.7, $s = 0.59$) with no trend over time. The intermediate pink variety could not always be distinguished from the purple but seems to vary around 1–3% (S5 Appendix).

We found direct effects of temperature and precipitation on flowering, as well as interactions between weather predictors and elevation for all three investigated periods (statistical analyses summarised in S8 Appendix). As exemplified in Fig 9, plots at relatively high elevations (i.e. plots 1–4) tended to have fewer flowers after a warm early spring (March–April) whereas plots at relatively low elevations (i.e. plots 5 and 6) tended to have more flowers when this period was warmer. The response to temperature was the same for the other periods (June and September in the previous year, Fig 10). For precipitation the reverse responses were seen: wet periods favoured flowering at high elevation and disfavoured flowering at low elevation (Fig 10).

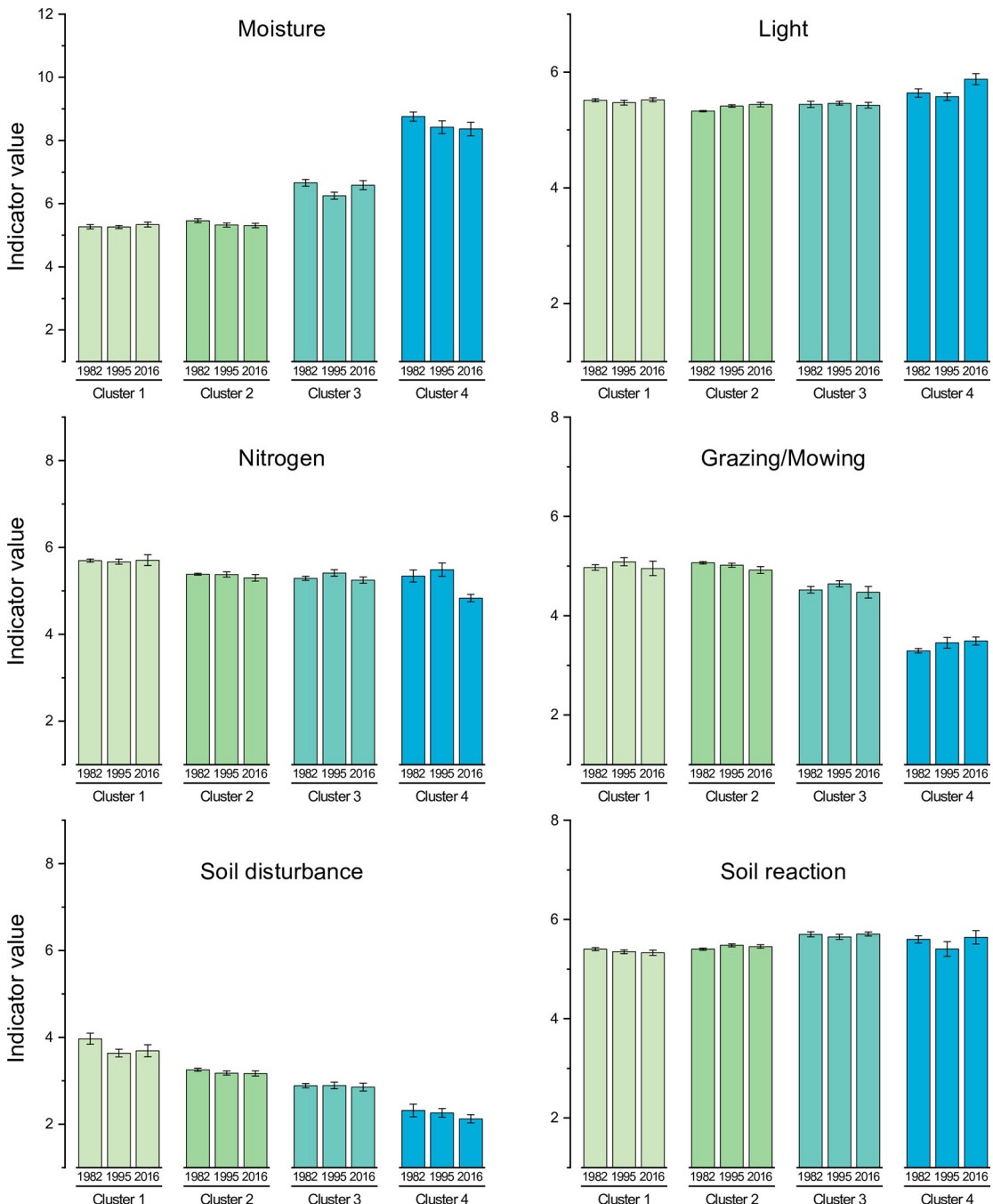

**Fig 6. Estimated marginal means ± 1 S.E. for abundance-weighted indicator values for plots in Profiles 1–4 regarding Moisture, Light, Nitrogen, Grazing/mowing, Soil disturbance and Soil reaction in 1982, 1995 and 2016.** Colours represent the four clusters with *N* values 12, 26, 21 and 17. The range of each *y*-axis covers the full range of the respective indicator. See S4 Appendix for mixed repeated-measures Anova.

There was a positive relationship between the number of flowers and the number of flowers in the previous year ($p = 0.019$ in GLMM; in this model there were no effects of elevation [$p = 0.30$] or any interaction [$p = 0.25$]).

The matrix analyses show that long-term population growth rates (λ) were 0.893 and 0.955 for the two yearly transitions. For both transitions the sensitivity matrices reveal that the most

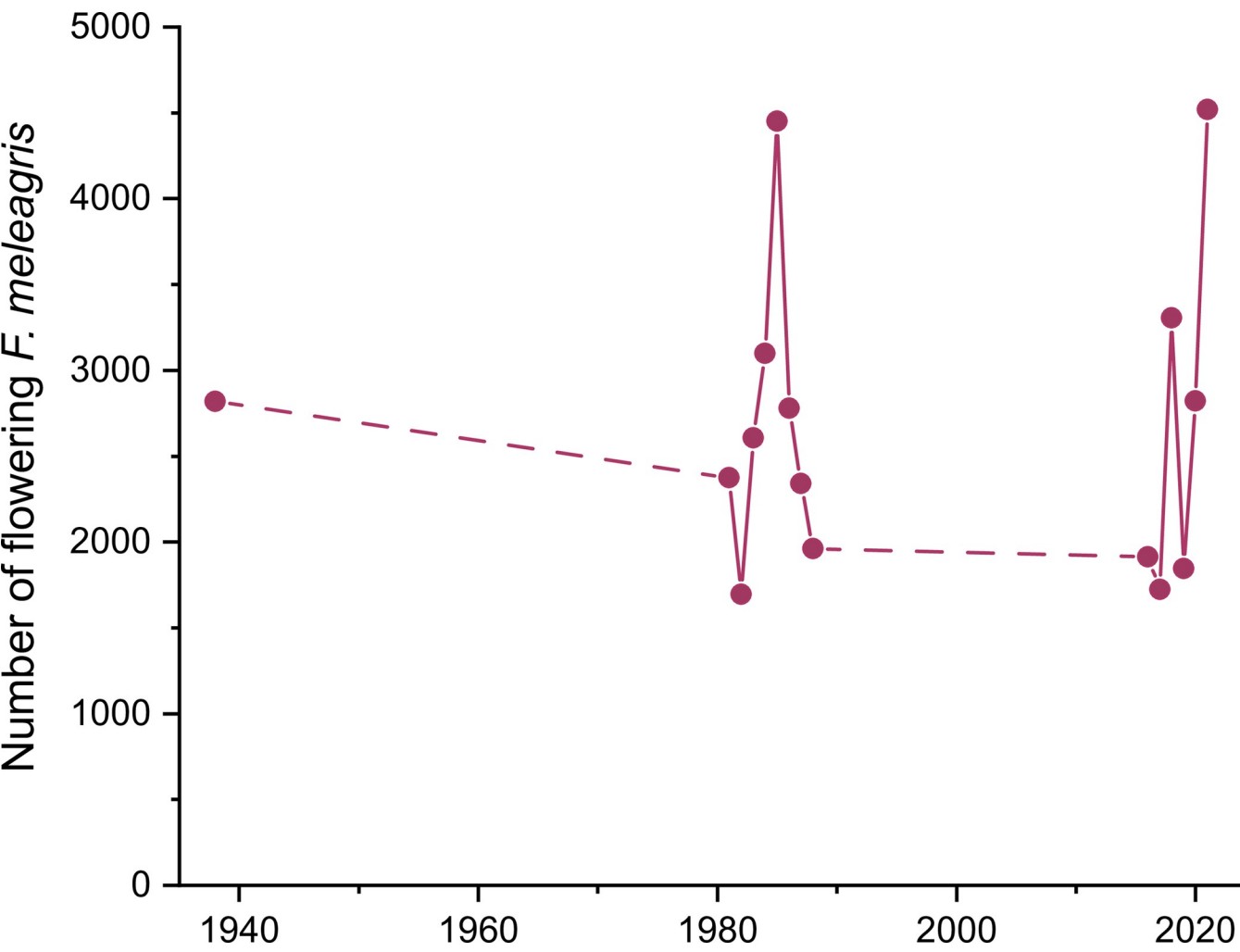

**Fig 7. Total number of flowering *Fritillaria meleagris* in six plots (10 m × 10 m) along Profile 1 (Fig 1) on Kungsängen, in 1938, 1981–1988 and 2016–2021.** Data in S5 Appendix.

important stages for long-term population growth rate were large vegetative plants surviving as large vegetative plants, becoming reproductive or retrogressing to medium-sized vegetative plants. Also, the survival of reproductive individuals was important, especially for the 1982–1983 transition. The contribution of the dormant stage for the 1981–1982 transition is small but noteworthy (see sensitivity analyses in S6 Appendix). Individuals entering dormancy were mainly from the smallest size class (83 of 196 and 77 of 128 individuals for the two transitions) and the opposite was also found, i.e. that most dormant plants entered the smallest size class (57% and 48%, respectively (see transition matrices in S6 Appendix).

## Discussion

### Changes in vegetation

Overall species richness and diversity at Kungsängen had not changed much since the 1940s. We had expected a decrease in species richness and diversity as the management has varied, with increased mechanisation, periods with too early mowing and grazing, and occasional tussock reduction. Also, the disappearance of the former surrounding meadows [12] could have

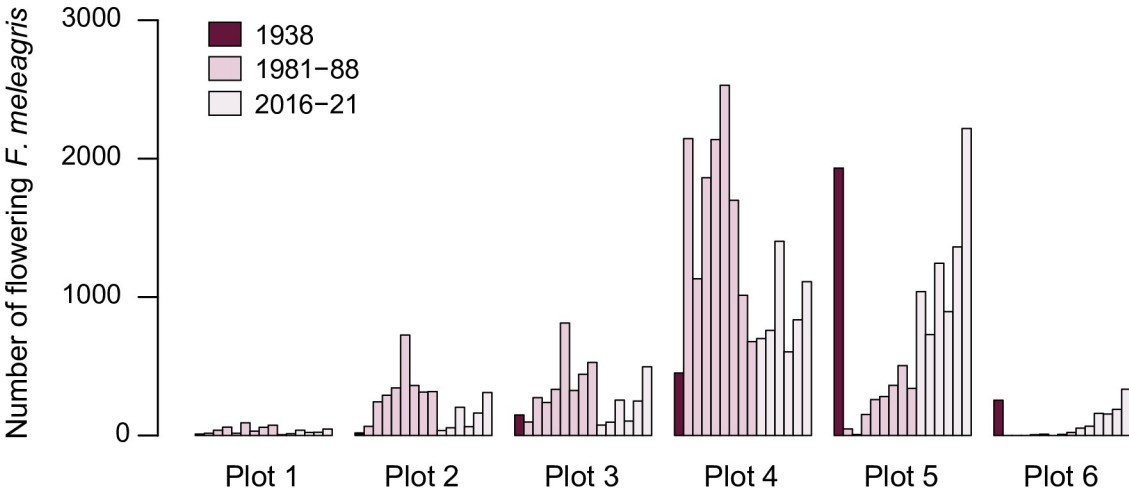

**Fig 8. Number of flowering *Fritillaria meleagris* in each of six plots (10 m × 10 m) along Profile 1 (Fig 1) on Kungsängen, in 1938, 1981–1988 and 2016–2021.** Plots 1–4 are in the drier part of the meadow (elevation ~1.8 m a.s.l.), and plots 5–6 in the wettest part of the range of *F. meleagris* (1.6 and 1.3 m a.s.l., respectively).

led to species loss as dispersal opportunities disappear [50]. The removal of hay should in the long run lead to a depletion of nutrients e.g. [51], but this could have been counteracted by airborne nitrogen deposition as well as flooding of the River Fyrisån over the wetter parts of the meadow.

In 1940, Sandberg distinguished four plant communities among the plots belonging to our Wet group. Even though his classification was subjective, it is obvious that the Wet part of Profile 1 had become more species-poor and very homogeneous between 1940 and 1982 (Fig 2A). As an example of the importance of spatial heterogeneity over the meadow, the decrease in species richness in the Wet part of Profile 1 between 1940 and 1982 was counteracted by an increase in the Mesic part, leading to an overall net increase in species richness and diversity. The increased diversity in the Mesic part is a result of the decreased dominance of *D. cespitosa* after 1940 (Fig 4A). Around 1940 Kungsängen was grazed from early on, rather than mown (S1 Appendix), a regimen that probably favoured *D. cespitosa* (cf. [52]).

The higher water level in the river after 1940 [18] would have led to more frequent flooding of the Wet part and partly explain the expansion of species such as *C. cespitosa*, *F. ulmaria*, and most notably, *C. acuta*. According to Sandberg [14], cutting *C. acuta* once during the vegetation season does not weaken it, and it is unattractive for cattle in the late season. However, the early and continued grazing practised around 1940 will restrict its distribution [14: p. 173]. For a few decades thereafter, management more or less adhered to the traditional late harvest followed by grazing. The conclusion is that the rise in water level in combination with late grazing from around 1950 was responsible for the expansion of *C. acuta*. Many studies have stressed the importance of management. For example, Rysiak et al. [53] noted a 60% decrease in species richness in abandoned plots compared to mown or grazed plots. Studies on calcareous pastures have shown the importance of continuous management, in this case grazing, for retaining a diverse and herb-rich vegetation [54]. It has also been shown that the vegetation could be rather robust to the timing of grazing if the grazing pressure is sufficient [55], but the expansion of *C. acuta* at Kungsängen indicates that timing is essential.

Contrary to our expectation, we found a decrease over time of the nitrogen indicator in Profile 1 (Figs 2 and 3). This was surprising, as the decrease would not be expected from the rise in water level in the river [18] since this would have increased flooding and nutrient

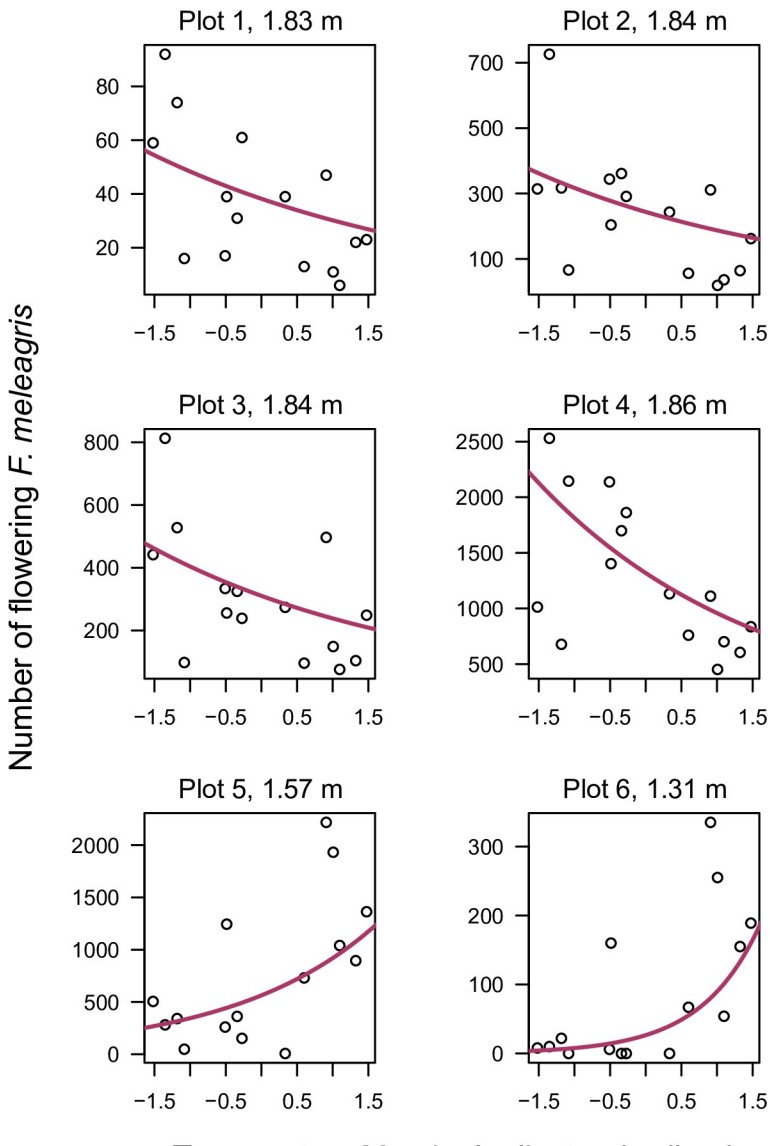

**Fig 9. Number of flowering *Fritillaria meleagris* in sixteen years between 1938 and 2021 in plots 1–6 along Profile 1 (Fig 1) in relation to elevation and mean temperature in March–April.** The lines show the number of flowers predicted by the generalised linear mixed model. Elevation (m a.s.l.) is given for each plot. Statistical analysis is summarised in S8 Appendix.

inflow, or the increased nitrogen deposition, but is consistent with nutrient depletion following intense management [56]. However, Krause et al. [57] could not detect any effects on floristics that could be attributed to nitrogen deposition in floodplain meadows in Germany. On the other hand, in a comparable study from England, Stevens et al. [58] found an increase in Ellenberg N scores from 1965 to 2012, which they attributed to increased atmospheric nitrogen deposition. Nitrogen effects are often confounded with effects of management and can be difficult to disentangle [59,60]. At Kungsängen, the grazing/mowing indicators decreased in the Wet part between 1940 and 1982, not supporting the hypothesis above regarding intense management. The answer may be found in the fact that mowing (no discrimination between

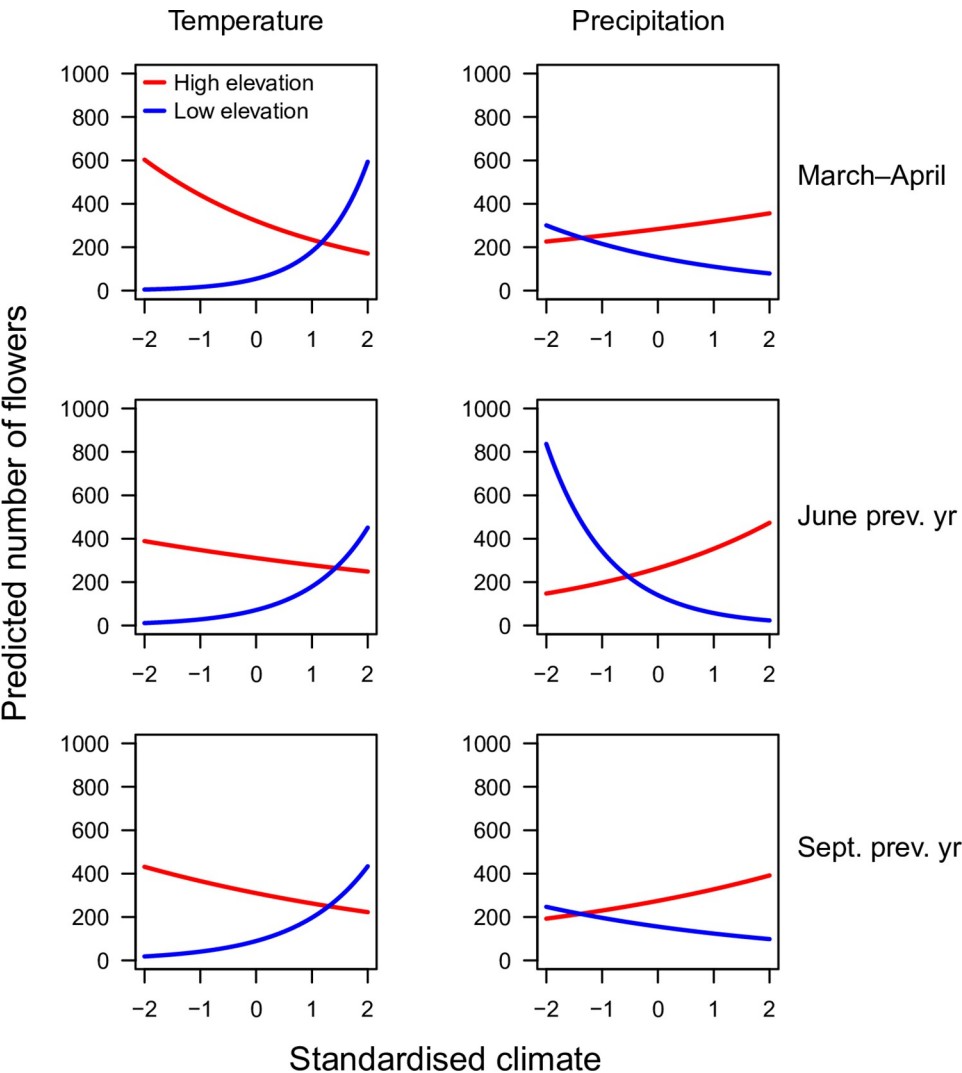

**Fig 10. Number of flowering *Fritillaria meleagris* in response to elevation and temperature or precipitation in the three active periods of the life cycle as predicted by generalised linear mixed models.** Red lines show predicted responses at the highest, and blue lines at the lowest elevation. Statistical analyses are summarised in S8 Appendix.

species) and grazing (often selective and leads to soil disturbance) have species-specific effects [61], and these two responses are confounded when they are merged into one indicator.

After 1982 we can consider community composition changes when analysing all four profiles, based on changes in the four clusters that line up along the first ordination axis in order of increasing wetness (Fig 5A). In addition, the position of Cluster 1, with its ruderal species, clearly reflects the effect of disturbance by tractor operation and by trampling of the cattle that gathered here. A barn in the easternmost part was removed several decades ago, and as a consequence, the ruderals and species richness have decreased and the vegetation has become more similar to Cluster 2. Some ruderals, such as the annual *Lithospermum arvense*, are still found close to the entrance, where the cattle gather. This is only a small area, but still important as it contributes to species richness and heterogeneity of the reserve.

The increasing wetness from Cluster 2 to 4 is coupled with decreasing indicators for Soil disturbance and Grazing/mowing. This may indicate that the, on average, taller wetland species at Kungsängen are less competitive under grazing or mowing than species in the more

mesic parts (cf. [62]. However, as discussed above, the Grazing/mowing indicator is difficult to interpret, since some species react differently to grazing compared to mowing [61].

In the wettest part of the meadow (Cluster 4), *C. acuta* dominated the vegetation with tall tussocks. Under a cutting regimen *C. acuta* will not form tussocks [63]. Without cutting and with periodic high water, tussocks are formed that provide micro-habitats for mesic meadow species [64], explaining why from 1995 a few meadow species are found in the wettest part of the meadow. In the ordination, Cluster 4 moved towards a drier species composition between 1982 and 1995 (Fig 5A) and became more heterogeneous with lower wetness indicator values and also more species rich. These changes did not continue after 1995, possibly counteracted by the rotor cultivator treatment in the late 1990s –the decline in *C. acuta* was smaller along Profile 1 where rotor cultivation was not applied than in the whole material (Fig 4).

The changes over time in Clusters 2 and 3 are smaller than in Clusters 1 and 4, and are more difficult to interpret. All four clusters have become more heterogeneous over time, but differences in indicator values are still consistently larger among clusters than among years. Our results, therefore, emphasise that habitat heterogeneity is particularly important in small nature reserves (cf. [65]): as we expected within-community changes have occurred in species richness and composition, but the current overall variation in the vegetation encapsulates the earlier variation, and more. While species richness declined in Cluster 1 and increased in Cluster 4 (for different reasons), overall species richness in 2016 was actually higher than in any previous survey. Thanks to the spatial heterogeneity, species richness appears resistant to variation in management.

The importance of continuous management for the maintenance of species richness in semi-natural grasslands is well known (e.g. [66]). Many studies have compared management methods, trying to ascertain the best method for maintaining species composition and diversity. A general conclusion from these studies is that confounding factors such as soil type, geographic location, land use history and species pool to a large degree influence the outcome (e.g. [67,68]). For example, Tälle et al. [69] in their meta-analysis found that the positive effects of grazing or mowing, respectively, depended on continent, grassland type and altitude, but when analysing the effects on plant species richness alone the effect of mowing and grazing did not differ. A combination of grazing and mowing may be best [53]. In the study by Gilhaus et al. [67], grazing was found to be beneficial for a high species richness; however, in their study the grasslands with a mowing regimen were more fertile, and in some cases even fertilised, a factor detrimental for species richness.

It has been suggested that the most beneficial treatment regimen is the one that has been in use historically [69], which, in the case for Kungsängen would be the traditional management with mowing in July–August (depending on plant phenology) followed by aftermath grazing. Continuous management in semi-natural grasslands does not mean, however, that there will be no changes in the vegetation. For example, Poptcheva et al. [70] reported succession and a slow immigration even after 20 years of mowing in a wet meadow. Furthermore, Köhler et al. [71], in an experiment with annual mowing for 22 years, noted that species identities and numbers were not constant and attributed that, besides immigration, to weather and different observers. Continued and even reinforced [72] management has been shown to be important for the maintenance of the often species-rich vegetation in semi-natural grasslands [73]. This was shown also by Köhler et al. [71], where all treatments involving weaker management than annual mowing resulted in reduced species numbers.

## Variation in flowering in *Fritillaria meleagris*

Given the huge yearly variation in the number of flowers and variation in management, the long-term stability is astonishing. In particular, we expected that the years with early harvests

could have been detrimental, but the numbers in 1938 and 2020 were almost the same: 2818 and 2822, respectively, and in the peak years (1985 and 2021) ca 4500 plants flowered. Zhang's [21] detailed demographic data indicate that most of the between-year variation in the number of flowers at Kungsängen is caused by variation in the propensity to flower rather than in population size, as the proportion of the population that flowered varied (23, 13 and 18%, for the years 1981–1983, respectively). Even larger variation in *F. meleagris* flowering was reported by Tatarenko et al. [24: data from Fig 6]: over 22 years the proportion that flowered varied between 10 and 59% at an English meadow, leading to a modest positive correlation between the number of flowering and vegetative plants (r ≈ 0.61). This is in line with observations of irregular flowering in other long-lived geophytes. For example, Pfeifer et al. [74] showed that in a steadily increasing population of the orchid *Himantoglossum hircinum* in Germany the number of flowering individuals increased at the same (logarithmic) rate as the total population, but with larger variation. Despite the large annual variation, the percentage of the *Fritillaria* population that flowers differed little between studies when averaged over years: 18% at Kungsängen, and 29% in England [24: data from Fig 6].

Our matrix population simulations based on the data collected by Zhang [21] indicate negative long-term population growth rates (λ < 1) in these years. However, the large difference in λ between the two transitions indicates that this result should be interpreted with caution. Also, plants assumed dead may, in fact, have been dormant as several years of prolonged dormancy have been observed [36] and has been suggested as a bet-hedging strategy [75]. If so, population growth rates would have been higher. We could not see any connection between flowering and prolonged dormancy as transitions to and from the dormant stage in our data mostly concerned the smallest stage class, as was found by Mehrhoff [76] and Calvo [77] for two North American orchids. This is in contrast to Lesica & Crone [78] who found for another iteroparous geophyte (*Silene spaldingii*) that dormant plants were more likely to flower in the following year. It is clear that a longer demographic study [79], incorporating mark–recapture statistics [80], would be necessary to get a clearer picture of how prolonged dormancy affects flowering and population growth.

Based on the sensitivity analysis we conclude that the most important stage for long-term population growth rates was the survival of large plants, whether vegetative or reproductive (S6 Appendix), as has been found for many other long-lived perennial plants (e.g. [27,81]). Even if reproduction by seed was not included in the simulations (seedlings could not reliably be distinguished in the field), one might assume that years of abundant flowering should also influence population growth rate positively and has contributed to the observed long-term stability of *F. meleagris* at Kungsängen. This is in contrast with other sites in the province of Uppland where the species has decreased slightly since the 1930s [8], probably as semi-natural grasslands have been abandoned (cf. [29,82]). However, *F. meleagris* has also dispersed to new sites [83].

The conclusion that the number of flowering individuals varies more than total population size led us to the assumption that the variability in flowering could be related to weather, for example by temperature affecting growth or soil moisture affecting aeration during the three active periods. *Fritillaria meleagris* has a restricted distribution along the wetness gradient [28,30], and at Kungsängen it mainly appears at an intermediate position along the wetness gradient (centrally placed in the species ordination plots; Figs 2B and 5B). The long-term stability of flowering could be explained by the ability of *F. meleagris* to "track" variation in wetness: wet and cold periods promoted flowering in the drier part of the meadow, and vice versa. In the long-term, the peak in flowering shifted from the wetter plot 5 to the drier plot 4 between 1940 and 1982 (Fig 8), consistent with the concurrent increased wetness indicated by vegetation changes in the wetter part in Profile 1 (related to the increased water level in the

river between 1940 and 1970 [18]). In England, Trist [28] similarly noted that there were more flowers of *F. meleagris* in the upper than in the lower part of a meadow in a wet year, and Cameron [30] found a strong negative correlation between number of flowers and the water table in the previous spring. At another English meadow, the highest number of flowers over a 20-yr period occurred after moisture conditions had been restored [29], in line with our results that wetness promoted flowering at the drier end of the niche of *F. meleagris* (Fig 10). Cameron [30] found that flowering was strongly promoted by a dry spring in the previous year ($R^2$ = 0.71 for number of flowers vs water level). We found that low precipitation in early summer in the previous year promoted flowering, but only in the wetter part of the meadow. Again, the small altitudinal difference of 30 cm between plots 4 and 5 caused contrasting responses. Similarly, in two English meadows (only 8 km apart) that were differently prone to flooding, flowering peaked in different years [24]. Together, these observations show that the way weather interacts with microhabitat conditions is more important than the direct effects of regional weather variation. In contrast to these opposing within-site responses to weather, Lindell et al. [27] found similar among-site responses to weather in *Pulsatilla vulgaris* ssp. *gotlandica*, which, however, has a more narrow wetness niche [84].

We cannot tease apart the effects of wetness and temperature. However, temperature should not vary much within the meadow, suggesting that wetness and soil aeration is important at several stages in the life-cycle before flowering. In a similar study, Tye et al. [85] analysed individual-based flowering data from 32 years for four orchid species in Norway. The study was conducted in mown fens, in which variation in soil aeration should play a similar role as at Kungsängen. They found that flowering was promoted by summer rain in the previous year in a coastal locality, and by summer temperature in an inland locality. However, the patterns were inconsistent, as one species showed no effect of weather, but our results show that responses to weather variation can go undetected if within-site spatial variation in microtopography is ignored.

We hypothesised that the propensity to flower could be related to the cost of earlier flowering, since a high reproductive effort in one year can reduce flowering in following years. The lack of a negative effect of flowering is surprising, since Zhang [21] showed that the size of the new bulb formed by flowering plants was considerably smaller than the ones formed by vegetative plants, and a small bulb would decrease the chances to flower in the next year. Even with long-term data, costs of reproduction is difficult to detect in natural populations [34], and the effects may differ among species and habitats: Tye et al. [86] removed flowers in two orchid species and in one species observed an increase in flowering the next year, followed by a decrease the year after, but a reversed pattern in the other species. The positive relationship we found with the number of flowering individuals in the previous year, could be a spurious relationship, since flowering is also related to weather in the previous year, as discussed above.

We hypothesised that the white morph should have increased since Kungsängen became a nature reserve and the preferential picking of the white morph stopped. But the variation has been, and still is, small and unchanging, around 3–4%, despite the large variation in the number of flowers. This is also supported by independent data collected at Kungsängen by Hedström [31]: for 1982 and 1983 he recorded 4.2 and 3.8% white flowers, respectively, and our result for the same two years was 3.5 and 4.2%, respectively. Zych et al. [32] similarly noted approximately 5% white flowers in a small anthropogenic population in N. Poland but reported only 10–30 white individuals in a large natural population (>>1 million) in S. Poland. According to Tatarenko et al. [24], British sites have in general <10% white flowers, but they also report a population with almost exclusively white flowers. The high percentage of white flowers in introduced populations is probably a founder effect caused by a local desire to breed the white morph. It seems that neither colour has a reproductive advantage [24]. At

Kungsängen, Hedström [31] found that pollinators (bumblebees, *Bombus* spp. and honeybee, *Apis mellifera*) show no colour preference, and there was no difference in the number of ovaries.

## Conclusions

In 1948 Rutger Sernander stressed the importance of the wetness zonation of Kungsängen, and the development over the last eight decades supports his view. Management and hydrology have unfortunately been poorly controlled and documented, but on the whole the spatial variation in wetness is intact.

The variation in management and the increased wetness in the wet parts between 1940 and 1980 have caused changes in different parts of the meadow. After 1982, species richness increased in the wettest part and decreased in the driest part. However, overall species richness and diversity have remained largely the same, suggesting that the vegetation is resilient to changes. This is true also for *F. meleagris*, which despite huge year-to-year variation showed long-term stability in the number of flowering individuals, the reason being that variation in temperature and precipitation acts in contrasting ways in the wetter and drier ends of the species' range.

Hence, species richness, overall species composition, and long-term stability in the *F. meleagris* population are maintained by the fine-scale variation in wetness, highlighting the importance of spatial heterogeneity and appropriate management as an insurance against biodiversity loss in semi-natural grasslands and nature reserves in general.

## Supporting information

**S1 Appendix. Hydrology and management at Uppsala Kungsäng.**
(PDF)

**S2 Appendix. Plot elevation and species cover in plots 1–76 in 1940–2016 (csv file [comma-delimited with decimal points]).**
(CSV)

**S3 Appendix. Anova of indicator values in Profile 1, 1938–2016.**
(PDF)

**S4 Appendix. Anova of indicator values in Profiles 1–4, 1982–2016.**
(PDF)

**S5 Appendix. Number of flowering *Fritillaria meleagris* at Uppsala Kungsäng.**
(PDF)

**S6 Appendix. Matrix analyses of *Fritillaria meleagris* 1981–1983.**
(PDF)

**S7 Appendix. Species with a significant change in cover between the first and last survey.**
(PDF)

**S8 Appendix. Generalised linear mixed models of the effects of climate and elevation on the number of *Fritillaria meleagris* flowers.**
(PDF)

## Acknowledgments

Mattias Vass and Brita Hytteborn assisted in the field. Robert Muscarella helped with statistical modelling and Scott Spellerberg revised the language.

## Author Contributions

**Conceptualization:** Håkan Hytteborn.

**Data curation:** Håkan Rydin.

**Formal analysis:** Håkan Hytteborn, Brita M. Svensson, Håkan Rydin.

**Investigation:** Håkan Hytteborn, Liquan Zhang, Håkan Rydin.

**Methodology:** Håkan Hytteborn, Bengt Å. Carlsson, Brita M. Svensson, Liquan Zhang, Håkan Rydin.

**Project administration:** Håkan Rydin.

**Writing – original draft:** Håkan Hytteborn, Bengt Å. Carlsson, Brita M. Svensson, Håkan Rydin.

**Writing – review & editing:** Håkan Hytteborn, Bengt Å. Carlsson, Brita M. Svensson, Liquan Zhang, Håkan Rydin.

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
