## [Decision Letter · Decision Letter 0]

28 Aug 2022

PONE-D-22-16087Zonation and spatial heterogeneity ensure long-term stability in vegetation and Fritillaria meleagris dynamics in the semi-natural grassland Uppsala KungsängPLOS ONE

Dear Dr. Rydin,

Thank you for submitting your manuscript to PLOS ONE. After careful consideration, we feel that it has merit but does not fully meet PLOS ONE’s publication criteria as it currently stands. Therefore, we invite you to submit a revised version of the manuscript that addresses the points raised during the review process. Both reviewers noted the study and data are valuable, nonetheless they have a long list of concerns and suggestions for manuscript improvement.  Notably, both identified flow and organization of the manuscript as problematic and needing significant modification.  Both also thought clarity would be significantly improved by focusing and better integrating concepts and aims, and reducing and streamlining text.

We look forward to receiving your revised manuscript.

Kind regards,

Dr. Janice L. Bossart

Academic Editor

PLOS ONE

https://journals.plos.org/plosone/s/file?id=ba62/PLOSOne_formatting_sample_title_authors_affiliations.pdf".

4. We note that [Figure 1] in your submission contain [map/satellite] images which may be copyrighted. All PLOS content is published under the Creative Commons Attribution License (CC BY 4.0), which means that the manuscript, images, and Supporting Information files will be freely available online, and any third party is permitted to access, download, copy, distribute, and use these materials in any way, even commercially, with proper attribution. For these reasons, we cannot publish previously copyrighted maps or satellite images created using proprietary data, such as Google software (Google Maps, Street View, and Earth). For more information, see our copyright guidelines: http://journals.plos.org/plosone/s/licenses-and-copyright.

Please upload the completed Content Permission Form or other proof of granted permissions as an """"Other"""" file with your submission.

Natural Earth (public domain): http://www.naturalearthdata.com/.

5. We note that you have referenced (Borgegård, S.-O. and Zhang, L. (1996) The vegetation at the Kungsängen nature reserve) which has currently not yet been accepted for publication. Please remove this from your References and amend this to state in the body of your manuscript: as detailed online in our guide for authors

Reviewers' comments:

Reviewer's Responses to Questions

**Comments to the Author**

1. Is the manuscript technically sound, and do the data support the conclusions?

Reviewer #1: Partly

Reviewer #2: Yes

2. Has the statistical analysis been performed appropriately and rigorously? 

Reviewer #1: Yes

Reviewer #2: I Don't Know

3. Have the authors made all data underlying the findings in their manuscript fully available?

Reviewer #1: Yes

Reviewer #2: Yes

4. Is the manuscript presented in an intelligible fashion and written in standard English?

Reviewer #1: Yes

Reviewer #2: No

5. Review Comments to the Author

Reviewer #1: The manuscript examines changes in species composition over time with respect to management at a wet to mesic semi-natural grassland in Sweden. The manuscript was interesting to read and utilises a valuable long-term dataset.

Although the study focuses on one survey site, long-term data is important for vegetation analysis, and the findings are important as they do not necessarily follow the trend of a loss in species diversity, which is evident on other semi-natural grasslands over time. However, I think the site-specific aspects of the manuscript could be toned down by reducing some of the text, as the manuscript is quite long in parts e.g. the discussion. The conclusion is very site specific and would benefit from a broader discussion. The flow of the manuscript could also be improved – at the moment it seems quite compartmental in term of vegetation results then the Fritillaria meleagris results, and also how the results from Profile 1 and profile 2-4 are presented. I appreciate they can’t be directly combined but the results could be more integrated. The method also jumps around from statistical analysis back to Fritillaria meleagris methods.

My main issue is the lack of flow from the aims to the discussion. I’m not sure whether the questions have been answered as they are not referred back to. I think the methods, results and discussion would be better structured in relation to the questions in the aims. This might help to reduce the word count and keep the manuscript more focused.

Please see specific comments below:

Introduction

L36: What are northern countries?

L54: e.g. not needed here

L138: You won’t be able to make conclusions regarding management based on differences between survey period alone. There could be other drivers influencing the vegetation during this time e.g. nitrogen deposition, so you’ll only be able to say changes might be due to management.

L143: Is this the only known species to be abundant since the 1750s?

L148: This seemed to come out of nowhere, it would be good to introduce this idea before the aims.

Figure 1. It would be useful to have an inset map indicating where the nature reserve is in Sweden and perhaps which part is a Natura 2000 site. I found it a bit confusing having the right-hand side plot here. I think this should be moved to the results and more explanation is needed e.g result of the ordination etc.

Method

L215: Perhaps emphasise the importance of relocation here for avoiding pseudo-turnover etc.

L233: It might be clearer to say Survey year.

L262-279: This information shouldn’t be in the methods. Move to the introduction?

L259: What is plot?

Figure 2: On the right-hand side, should this say profiles 2-4? Label the plots A and B would be clearer.

L286: How are you relating number of flowers to temperature? What method are you using?

Results

L311 and onwards: It is better to discuss p-values as a “notion of evidence” rather than significant or not. See paper by Muff et al. (2022) TREE.

Figure 3: Include (n = ) for plots. You haven’t described the eclipses? It would be good to describe the function you used to make these in the methods. Also I don’t remember much discussion in the text for the inset.

L325: It would be good to label the most abundant species on ordination in Figure 3 and then include all species in the Supp Mat as you have done already.

L331: Shifted down on NDMS axis 2?

Figure 4: The colours used in 4a are not colour blind friendly. Can the points be made any larger, I found it quite hard to see.

L338: Why have you just looked at 1982? I’m not sure how these four clusters have been classified.

L343: Give some examples of the types of ruderal plants

L345: Would you expect Fritillaria meleagris to be positioned here?

L386: Again, see earlier comment about notion for evidence rather than “statistically significant differences”.

L396: Should this be Profiles 2-4?

Figure 7. You need more than just “number” on your y -axis.

L418: None of these could predict the number of flowers? How did you determine this? Where are the results for this?

L428: Again this part seems like its tacked on at the end.

Discussion

This section was very long and could be cut down. The discussion didn’t seem to link back to the original aims, so I am unsure on whether the questions have been answered. In particular, there didn’t seem to be a link between Fritillaria meleagris and the vegetation changes? Is Fritillaria meleagris a good indicator for examining vegetation change? It might be worth looking at this paper: Stroh et al. (2017) Plant Ecology

L438: More discussion on nitrogen deposition would be useful here.

L448: Does?

L468: Are there any other studies which have found a similar thing with Deschampsia?

L573: The weather effect was only found when using 2 plots?

Conclusions and implications for management

This section was very site specific, it would be nice to try and relate more general conclusions to semi-natural grasslands. I think it’s worth emphasising that despite the changes in management, the vegetation remained largely consistent, suggesting it is robust to change. See Ridding et al. 2021 Journal for Nature Conservation for similar results.

Reviewer #2: Topographical heterogeneity of floodplains is important factor for sustaining species-rich vegetation. Hydrological gradient ensures the species with different water requirements to flourish in dynamic floodplain environment under annual climatic variations. The difference in elevations as small as 10 cm often results in vegetation changes.

This paper describes changes in vegetation of Kungsäng lowland meadow over 80 years in relation to the spatial heterogeneity of the site. The enormous wealth of data, partly already published, partly new, along with a substantial expertise developed in several periodic studies of this particular site were put together in this paper. Diverse information, which is not always relevant to the topic (e.g., pollination of Fritillaria meleagris or color variations of its flowers, etc, etc) has been pulled into one paper. As a result, the main idea of spatial heterogeneity (=hydrological gradient? =topographical zonation?) influencing vegetation diversity and population dynamics, is sometimes difficult to follow.

Spatial heterogeneity is not clearly defined and shown on the map (Fig.1). Importance of micro-topography is only stated towards the end of the discussion, but not mentioned earlier in the paper. Showing elevations in this large and complex wetland would help to define “zonation” announced in the title, and make presentation and interpretation of the data much clearer. Vegetation units, like “wet” and “mesic”, or four undefined “clusters” are not explained. Why two of these approaches are required? How do they add to each other? Would it be possible to combine all data from all surveys into new analysis based on the plots elevations? If positions of plots along the profiles can be accurately located, they can be easily linked to the topographic elevation values. Such allocation along the elevation gradient would link vegetation data to the physical properties of the site and justify “zonation” idea both in analysis and interpretation of the results. Currently wetness of the site is measured by the river levels from the distant gauge boards. Those values are the same for entire site. Elevation is only a proxy which can be applied to individual plots along all four profiles (data are freely available from LiDAR system) and reflect variation in wetness across the site.

Management, which is undoubtedly one of the major drivers of vegetation composition, presented in this paper in a very descriptive way, which is difficult to follow. It needs to be formalised and linked clearly to both time line of the data collection and to the spatial heterogeneity of the site. Management is notoriously difficult to quantify and put into the data analysis. However, it is still possible as categorical data. If analysis is not possible, a role of management should be only mentioned in Introduction and in Discussion, in general terms. It can’t be outlined in the Conclusion as effects of management on vegetation were not studied in this paper.

The second large aspect of this study is population analysis of Fritillaria meleagris, as one of the iconic species of Kungsängen. It would be most interesting to see the link between

the flowering dynamics of Fritillaria meleagris and vegetation changes studied in the same observation plots. If such analysis could’ve been done, the paper looked more uniform and linked. Again, population data analysed against elevation gradient would answer the question about effect of microtopography on population dynamics.

Following the title of the paper, data analysis of the Fritillary population should be focused on spatial heterogeneity of the site. Differences in elevations between 6 observation plots should be quantified. Why and how only two out of six plots have been selected for a detailed analysis of effect of “wetness” ought to be clearly explained in Methods. Conclusion about Fritillary “wet” and “mesic” population’s loci reacting differently to the variation of the soil moisture in different seasons sounds very intriguing even correlations shown in Fig.9 are not strong. This observation deserves more focused discussion, based on authors’ knowledge of seasonal species growth.

Referencing in this paper is very dense and intense. References occur even in Results where the findings of the current paper are presented. Very many quotes include the page number or the Figure number from other publications. That implies, the reader of the current paper has to keep at least a dozen of other publications opened, in order to follow the logic. References are not formatted according to Plos One requirement.

The paper can be published after a major revision and re-writing. English language should be checked by native English speakers. In many places the writing is too descriptive. Numerous self-references to the previous most interesting and valuable publications of some authors, often overload the paper with details which are not directly relevant to the topic of this paper.

The title:

“Zonation” is a too general, not very explanatory term. It always require an adjective saying which factor has been “zoned”. Hydrology? Soil compaction? Soil mechanical composition? Grazing?

What is a difference/link between “zonation” and “spatial heterogeneity”?

There is no description/definition of the zones in the paper.

Line 22: “Vegetation clusters” is not a well known term, needs clarification concerning a principle point of clustering.

Line 25: “affected by weather”. It needs to be more specific about the timing of weather. Long-term? Previous year? Spring of the same year?

Line 50: broad-leaved grasses… What does it mean?

Line 52: litter is a part of biomass. Directly, neither of them affects seed germination. Thick litter cut off an access to the ground, that should be explained.

Line 57: flooded habitats are not an “exception from the rules”. Hay cut still remove nutrients there. The sentence needs a different wording.

Line 66: if Glyceria maxima is exotic to Swedish flora, that must be stated. Otherwise, this is a common aquatic species. Was an appearance of the species in the nature reserve caused by particular management?

Line 79: Vice versa – is not clear in the given context.

Line 101: Is “lily” a Swedish short name for Fritillary? If yes, that should be mentioned in the line 99. English name is snake’s head fritillary.

Lines 123-124: the statement about perennial plants is not clear, needs re-writing. Why the allocation of resources in previous year reduces flowering?

Line 130: probability of individual flowering was estimated by Tatarenko (2019). Useful paper about Fritillary populations.

Lines 157-159: Is meadow situated on the floodplain? When it was at the sea level, was there floods from the sea? Not sure how the lake 6 km downstream influences the water table in the meadow. Were there any hydrological observations/measurements ever done?

Line 183: “permanent annual mowing regime”? Either permanent or annual… Regular?

Lines 150-208: Location of the site should be described in Introduction, not in the Methods. Description of hydrology and management of the site is presented as a literature review, not as “Methods”. The information is very descriptive and difficult to follow to a reader who is not familiar with the site. If the data from the quoted publications are used in the vegetation and population analyses in this paper, they should be presented in the Table, separately for hydrology and especially, for management. Elements of Literature review should be moved to Introduction, where the site is described in details.

The map of the site (Fig. 1) needs showing “zonation”, both hydrological and management-wise.

The same applies to Vegetation “Methods”: having a Table showing which plots in which profile were surveyed in which year and by which botanist would allow much better understanding of the structure of botanical data.

Vegetation clusters – a non-conventional approach to vegetation classification. It must be described in details in Methods. What are the criteria for definition of the clusters?

Why two approaches of grouping the species in vegetation units were needed? One is “wet” and “mesic”, another – four clusters? Why a traditional vegetation classification was not used?

Lines 261-279 – again, this is not the Methods, this is a literature review on the species biology. This should be placed in Introduction. Some details (e.g., pollination, seasonal growth) are not relevant to the topic discussed in the paper.

Lines 288-289 – the reference is a repeat of the same above in Line 163.

Line 307: give values for “highest”

Line 312-313: Such prolonged gaps between observations don’t support conclusions about “directional” changes in this extremely dynamic vegetation.

Line 318: Changes in Vegetation. Would it be a better title? “Development”, as well as “succession” would mean one-directional changes to a higher organisational level.

Line 331: What is a factor which drives communities “downwards”? The title of the paper is “zonation and spatial heterogeneity…” How does the spatial heterogeneity apply to the results described here?

Lines 349-380: changes in four clusters are presented in a very descriptive way, which is difficult to follow. Table 1 gives values of increase / decrease in species cover recorded in four surveys. As the main focus of the paper is the “spatial heterogeneity” (which is represented by “clusters” to large extent, isn’t it?), it would be more informative to group changes in species abundance according to the “clusters”.

Line 383-388: levels of significance should be shown.

Line 383-384: “higher” than what?

Line 421: “precipitation” is mentioned here for the first time. All weather factors should be mentioned in Methods. Was precipitation actually measured/analysed?

Lines 433-465: this information belongs to Introduction, no discussion of the findings of the current paper here.

Line 466: Variations in management against changes in species richness should be first presented in the Results section. That would justify discussion presented here.

Line 479: “zonation” was not mentioned in Results

Lines 529-534: very difficult to follow logic here

Line 536: “root cultivation” needs to be explained

Line 546-547: the question about “dynamics of population or”… is not clear.

Line 548-557: how does it link to the topic of the paper?

Line 555: purpose for re-analysis of Zang (1983) dataset should be explained in Introduction. As supplementary material explains, only dormancy has been added to the published analysis. Why that was required? No new data were collected recently. Assumption of plants being dead after 2 years of not being seen above the ground is not justified. Two-year dormancy in Liliaceae and Orchidaceae is as common and one-year dormancy. This analysis doesn’t add any new interpretation of the plant distribution/dynamics along the hydrological gradient.

If a use of this analysis had been justified in Introduction, model should be explained in Methods, or reference to the Supplementary material given there. Data from the analysis should be shown in the Results, of reference to Supplementary material given there. After that it can be discussed here.

Line 563-564: Not quite correct extraction of information from Tatarenko et al 2022. Fig 6 there doesn’t show “flowering and vegetative adults”. It shows ratio between flowering and vegetative plants, not “vegetative adults”.

Line 558-571: How does it relate to the topic of the paper?

Line 578: How exactly do fritillary plants “track long-term changes in wetness”?

Line 604-616: mainly repeat of old data and a small and non-conclusive finding from the current project discussed against the data for not taxonomically close species. This doesn’t add much to the main story discussed in the paper.

Line 617-628: not relevant to the topic of the current paper.

6. PLOS authors have the option to publish the peer review history of their article (what does this mean?). If published, this will include your full peer review and any attached files.

Reviewer #1: No

Reviewer #2: No

---

## [Author Response · Author response to Decision Letter 0]

22 Nov 2022

Dear Editor,

We now submit a revised version of the manuscript PONE-D-22-16087, now entitled 'Spatial heterogeneity ensures long-term stability in vegetation and Fritillaria meleagris flowering in Uppsala Kungsäng, a semi-natural meadow'. We summarize our changes in the response to the Editor's comment (next section), and then respond to all comments from the two reviewers. 

PONE-D-22-16087

Zonation and spatial heterogeneity ensure long-term stability in vegetation and Fritillaria meleagris dynamics in the semi-natural grassland Uppsala Kungsäng

PLOS ONE

Dear Dr. Rydin,

Thank you for submitting your manuscript to PLOS ONE. After careful consideration, we feel that it has merit but does not fully meet PLOS ONE’s publication criteria as it currently stands. Therefore, we invite you to submit a revised version of the manuscript that addresses the points raised during the review process.

Both reviewers noted the study and data are valuable, nonetheless they have a long list of concerns and suggestions for manuscript improvement. Notably, both identified flow and organization of the manuscript as problematic and needing significant modification. Both also thought clarity would be significantly improved by focusing and better integrating concepts and aims, and reducing and streamlining text.

Response: We really appreciate the constructive comments from the reviewers, that forced us to re-think and re-analyse our results. We have described the responses in detail below, and the important overall changes are:

1. The previous presentation of vegetation dealt too much with the overall changes in the meadow. As the reviewers noted, the focus should be on spatial heterogeneity, and we replaced presentations that covered the whole profiles (previous Fig. 2 deleted, and Table 1 moved to Appendix S7) with the new Fig. 4 that describe changes in different parts of the meadow. From this follows that the Discussion also focuses on the spatial differences,

2. We followed the suggestion from Reviewer 2 to include LiDAR elevation data to be able to explicitly describe the spatial heterogeneity.

3. With the LiDAR data we now model the number of Fritillaria flowers based on all plots with generalised linear mixed models (GLMM) to evaluate the interactive effects of climatic variation and elevation. 

4. Thanks to these changes the links between vegetation and Fritillaria dynamics became more obvious.

5. Text streamlining includes clarifications of aims, clear links to aims in Discussion, transfer of descriptive parts from Methods to Introduction (as suggested by Reviewer 2), and of some background material to Appendix S1. 

Journal requirements

Response: Done

Response: The statement is now: ‘Data are available in Appendices S2 and S5. Weather data for Ultuna weather station are available at https://www.ffe.slu.se/lm. Species indicator values are available in: Tyler T, Herbertsson L, Olofsson J, Olsson PA. Ecological indicator and trait values for Swedish vascular plants. Ecol Indic 2021;120: 106923. https://doi.org/10.1016/j.ecolind.2020.106923.’

Response: We have added the statement that we have permission from authorities to work in the nature reserve. 

4. We note that [Figure 1] in your submission contain [map/satellite] images which may be copyrighted. All PLOS content is published under the Creative Commons Attribution License (CC BY 4.0), which means that the manuscript, images, and Supporting Information files will be freely available online, and any third party is permitted to access, download, copy, distribute, and use these materials in any way, even commercially, with proper attribution. For these reasons, we cannot publish previously copyrighted maps or satellite images created using proprietary data, such as Google software (Google Maps, Street View, and Earth). For more information, see our copyright guidelines: http://journals.plos.org/plosone/s/licenses-and-copyright.

Response: Response: This image is deleted. Instead we made a map showing variation in elevation in the meadow as suggested by Reviewer 2, a suggestion that we really appreciated!

5. We note that you have referenced (Borgegård, S.-O. and Zhang, L. (1996) The vegetation at the Kungsängen nature reserve) which has currently not yet been accepted for publication. Please remove this from your References and amend this to state in the body of your manuscript: as detailed online in our guide for authors

Response: Done

Reviewer's Responses to Questions

Comments to the Author

1. Is the manuscript technically sound, and do the data support the conclusions?

Reviewer #1: Partly

Reviewer #2: Yes

2. Has the statistical analysis been performed appropriately and rigorously? 

Reviewer #1: Yes

Reviewer #2: I Don't Know

3. Have the authors made all data underlying the findings in their manuscript fully available?

Reviewer #1: Yes

Reviewer #2: Yes

4. Is the manuscript presented in an intelligible fashion and written in standard English?

Reviewer #1: Yes

Reviewer #2: No

Response: The text is now checked by a native English speaking professional proofreader.

Review Comments to the Author

Reviewer #1 

The manuscript examines changes in species composition over time with respect to management at a wet to mesic semi-natural grassland in Sweden. The manuscript was interesting to read and utilises a valuable long-term dataset.

Although the study focuses on one survey site, long-term data is important for vegetation analysis, and the findings are important as they do not necessarily follow the trend of a loss in species diversity, which is evident on other semi-natural grasslands over time. However, I think the site-specific aspects of the manuscript could be toned down by reducing some of the text, as the manuscript is quite long in parts e.g. the discussion. The conclusion is very site specific and would benefit from a broader discussion. The flow of the manuscript could also be improved – at the moment it seems quite compartmental in term of vegetation results then the Fritillaria meleagris results, and also how the results from Profile 1 and profile 2-4 are presented. I appreciate they can’t be directly combined but the results could be more integrated. The method also jumps around from statistical analysis back to Fritillaria meleagris methods.

Response: We have tried to fulfil what the reviewer is asking for. The text is clearer; in the Discussion we have integrated more references to other studies to make conclusions less site-specific; we present better links between the Fritillaria and vegetation parts; the Methods have been restructured as suggested. Some background details are moved to Appendix S1. Details about the changes are given below in response to specific comments. 

My main issue is the lack of flow from the aims to the discussion. I’m not sure whether the questions have been answered as they are not referred back to. I think the methods, results and discussion would be better structured in relation to the questions in the aims. This might help to reduce the word count and keep the manuscript more focused.

Response: We have tried hard to improve structure along these lines. This includes clarifying the aims in the introduction and referring back to them in the discussion

Please see specific comments below:

Introduction

L36: What are northern countries?

Response: Changed to ‘large parts of Europe’.

L54: e.g. not needed here

Response: Removed.

L138: You won’t be able to make conclusions regarding management based on differences between survey period alone. There could be other drivers influencing the vegetation during this time e.g. nitrogen deposition, so you’ll only be able to say changes might be due to management.

Response: We have toned this down and incorporated the potential role of other drivers in the aim in the following formulation (line 138): 'We will view the vegetation composition found in the 1940 survey as a reflection of traditional management of a wet–mesic meadow. Since then, the management has periodically adopted the modern practice of early hay harvest, as well as being mechanised, and we hypothesise that this (in combination with other environmental changes, such as nitrogen deposition) may have led to losses in species richness and plant diversity and to changes in community composition.'

 We have added the information that nitrogen deposition peaked in the 1990s. However, the nitrogen indicator had decreased between 1940 and 1982, so we see no clear evidence for an effect of N deposition. Still, we agree with the reviewer that it is difficult to make conclusions about the exact role of management. But in the text we conclude that despite variation in management and other factors, species richness has not declined, and the number of flowering Fritillaria shows no long-term trend (although it varies enormously from year to year). 

L143: Is this the only known species to be abundant since the 1750s?

Response: Linnaeus did not give indications of abundance of other species, but we now mention some sea-shore plants that were found at the meadow by Linnaeus: Glaux maritima and Triglochin maritima (have disappeared) and Alopecurus arundinaceus (still present).

L148: This seemed to come out of nowhere, it would be good to introduce this idea before the aims.

Response: We don't agree that this came out of nowhere; the colour variation was introduced before aims (with the statement that the white morph was preferred by flower-picking townsfolk). The description of Fritillaria biology (with information about flower colours) is now moved from Material and Methods to Introduction. This particular research question is also now more clearly outlined in the Introduction and referred back to in the Discussion ('We hypothesised that the white morph should have increased...').

Figure 1. It would be useful to have an inset map indicating where the nature reserve is in Sweden and perhaps which part is a Natura 2000 site. I found it a bit confusing having the right-hand side plot here. I think this should be moved to the results and more explanation is needed e.g result of the ordination etc.

Response: We added the inset map. The map now contains elements of site description, study design and spatial heterogeneity and as Reviewer 2 urged that ‘Location of the site should be described in Introduction’ we first refer to the map in the Introduction. We deleted the Natura 2000 reference (not particularly relevant). Based on a suggestion from Reviewer 2 plot elevation has become an important part of the analyses, and the new figure illustrates elevation variation over the meadow. The information from the previous right-hand map has now been superimposed on the elevation map.

Method

L215: Perhaps emphasise the importance of relocation here for avoiding pseudo-turnover etc.

Response: Even with a fairly precise relocation of plots some pseudo-turnover is inevitable. The risk of pseudo-turnover is one reason why we focus our analyses on groups or clusters of plots rather than individual plots.

L233: It might be clearer to say Survey year.

Response: Changed to Survey year throughout

L262-279: This information shouldn’t be in the methods. Move to the introduction?

Response: Moved to Introduction as suggested. 

L259: What is plot?

Response: Changed to Plot identity.

Figure 2: On the right-hand side, should this say profiles 2-4? Label the plots A and B would be clearer.

Response: 'Profiles 1-4' was correct. In 1940, only Profile 1 existed. Profiles 2-4 were added in 1982, so from 1982 we could analyse all profiles. However, this figure is replaced by the new Fig. 4 (which now shows what’s happening in the different clusters and labelled as such). So, in contrast to the previous Fig. 2, the new figure shows changes in different parts of the meadow. This helps to highlight the spatial heterogeneity which is our focus. 

L286: How are you relating number of flowers to temperature? What method are you using?

Response: This is now clarified in the Analyses section. In the previous version we used linear regression. Following suggestions by Reviewer 2 we now use non-linear mixed models (GLMM) combining weather (temperature, precipitation) and plot elevation as predictors.

Results

L311 and onwards: It is better to discuss p-values as a “notion of evidence” rather than significant or not. See paper by Muff et al. (2022) TREE.

Response: As described in the paper by Muff, this is a century-long debate among statisticians seemingly without agreement. We don't feel competent to evaluate this or brave enough to completely leave the tradition with preset alpha values. However, for the important analyses on which we base our argument we provide p-values to allow the reader to judge (Appendices S3, S4, S8). We use the 'notion of evidence' terminology in reference to Appendix S7 (previously Table 1) where we kept the threshold p < 0.05 to select species to be tabulated.

Figure 3: Include (n = ) for plots. You haven’t described the eclipses? It would be good to describe the function you used to make these in the methods. Also I don’t remember much discussion in the text for the inset.

Response: In the legend to Figs 2a and 5a we have added n-values. We added in Methods that we show convex envelopes (with reference to the ‘ordihull’ function in vegan). The inset (Fig. 2a) is now presented as part of the results from analysis of indicator values. This should be more clear now as the changes in vegetation and indicator values are presented under the same heading in Results (not separately as in the previous version). 

L325: It would be good to label the most abundant species on ordination in Figure 3 and then include all species in the Supp Mat as you have done already.

Response: As suggested we have added species plots to both ordinations (Figs 2a and 5a) including the most abundant species, and also other relevant species that are mentioned in the text (e.g. some ruderals). To improve the link between vegetation and Fritillaria, we now highlight the position of Fritillaria in these plots. 

L331: Shifted down on NDMS axis 2?

Response: Changed as suggested.

Figure 4: The colours used in 4a are not colour blind friendly. Can the points be made any larger, I found it quite hard to see.

Response: This is now Fig. 5a. One of us is colour blind and thought it was OK, but we have tried to make the graph easier to read by altering the colour code and making points larger. 

L338: Why have you just looked at 1982? I’m not sure how these four clusters have been classified.

Response: The ordination was made for all years (1982-2016), but we now clarify that we use the clusters in 1982 to describe the starting point, since 1982 was the first year for analyses that included all four profiles. Then we could follow what happened in the different parts of the meadow, defined by these 'starting' clusters. (Similarly we used the wet and mesic groups in 1940 to describe the starting point for the analyses that included Profile 1 only). 

 In Methods we have now given details of the clustering methods. In the previous version we showed that the clusters were spatially separated, and now we have also added how the clusters relate to variation in elevation over the meadow (with the addition of elevation data), and here in Results (line 322) we describe that the four clusters were '(i) spatially separated in the meadow (Fig. 1), (ii) well separated in the ordination described below (Fig. 4) and (iii) reflect the elevation gradient (mean elevation for Clusters 1-4 was 2.0, 1.9, 1.5 and 1.3 m a.s.l.).'

L343: Give some examples of the types of ruderal plants

Response: Capsella bursa-pastoris and Poa annua are added as examples. 

L345: Would you expect Fritillaria meleagris to be positioned here?

Response: Yes. Most of the occurrences are in the Fritillaria plots 4 and 5 and these are within Cluster 2 and 3 along Profile 1. There is very little Fritillaria in Clusters 1 and 4, so it is to be expected that Fritillaria is centrally positioned in the species plot of the ordination. We note this in the Discussion (line 573): 'Fritillaria meleagris has a restricted distribution along the wetness gradient [31,33], and at Kungsängen it mainly appears at an intermediate position along the wetness gradient (centrally placed in the species ordination plots; Figs 2b and 5b).'

L386: Again, see earlier comment about notion for evidence rather than “statistically significant differences”.

Response: See response to L 311 above

L396: Should this be Profiles 2-4?

Response: No, the analyses from 1982 include all four profiles (1-4), the analyses from 1940 include only Profile 1. 

Figure 7. You need more than just “number” on your y -axis.

Response: Changed to 'Number of flowering F. meleagris', also in Fig. 8

L418: None of these could predict the number of flowers? How did you determine this? Where are the results for this?

Response: We have re-analysed the data using GLMMs with temperature or precipitation and elevation as predictors. As focus is on spatial heterogeneity, the main interest is then on the interaction between elevation and weather predictors (Figs. 9-10).

L428: Again this part seems like its tacked on at the end.

Response: See response to comment at L 148.

Discussion

This section was very long and could be cut down. The discussion didn’t seem to link back to the original aims, so I am unsure on whether the questions have been answered. In particular, there didn’t seem to be a link between Fritillaria meleagris and the vegetation changes? Is Fritillaria meleagris a good indicator for examining vegetation change? It might be worth looking at this paper: Stroh et al. (2017) Plant Ecology

Response: We have now tried to link back to the aims, and the aims are now more clearly stated. We appreciate the request for a better link between the Fritillaria and vegetation results. Fritillaria may not be a good indicator for vegetation changes, but both Fritillaria and vegetation are used to examine changes. Since the vegetation changes are now discussed in separate parts of the meadow (rather than as overall changes), it is easier to see the links. For example, vegetation data suggest that the wetter parts of Profile 1 became wetter between 1940 and 1982 (Fig. 2a). This could be linked with the observation that peak flowering moved upwards from Plot 5 to Plot 4 during the time (Fig. 1), and this is linked to the mechanism that wet periods disfavour flowering Fritillaria at the wetter end of its range (Fig. 10). This is outlined in lines 576-584 in the new version. We also highlight the position of Fritillaria in the vegetation analyses (Fig. 2b and 5b). The paper by Stroh is relevant and is now cited. 

L438: More discussion on nitrogen deposition would be useful here.

Response: We have added information about how nitrogen deposition has varied (peaked in the 1990s), and discuss its possible role in the changes. In essence we conclude that the nitrogen indicator values decreased during the time that nitrogen deposition increased, which suggests that changes in nitrogen availability has not been an important driver for vegetation changes.

L448: Does?

Response: Text changed, so no longer relevant.

L468: Are there any other studies which have found a similar thing with Deschampsia?

Response: We added a reference to Pruchniewicz (2017), indicating that the high cover of Deschampsia in 1940 was associated with weak mowing, and relate this to the known fact that around 1940 Kungsängen was grazed early (and hence not mown). 

L573: The weather effect was only found when using 2 plots?

Response: Thanks to the suggestion by Reviewer 2 to include elevation data for all plots as predictor we now present GLMM analyses that include all six plots. The new results are much more powerful and general (including all plots), but the results lead to the same conclusions. 

Conclusions and implications for management

This section was very site specific, it would be nice to try and relate more general conclusions to semi-natural grasslands. I think it’s worth emphasising that despite the changes in management, the vegetation remained largely consistent, suggesting it is robust to change. See Ridding et al. 2021 Journal for Nature Conservation for similar results.

Response: The robustness is also our main conclusion, and we have tried to stress it more now. We added the suggested reference in the Discussion where we also relate to other references to make the text more general. As always, this is not easy, since each site has its peculiarities. 

Reviewer #2

Topographical heterogeneity of floodplains is important factor for sustaining species-rich vegetation. Hydrological gradient ensures the species with different water requirements to flourish in dynamic floodplain environment under annual climatic variations. The difference in elevations as small as 10 cm often results in vegetation changes.

This paper describes changes in vegetation of Kungsäng lowland meadow over 80 years in relation to the spatial heterogeneity of the site. The enormous wealth of data, partly already published, partly new, along with a substantial expertise developed in several periodic studies of this particular site were put together in this paper. Diverse information, which is not always relevant to the topic (e.g., pollination of Fritillaria meleagris or color variations of its flowers, etc, etc) has been pulled into one paper. As a result, the main idea of spatial heterogeneity (=hydrological gradient? =topographical zonation?) influencing vegetation diversity and population dynamics, is sometimes difficult to follow.

Response: We have tried throughout to clarify and more clearly focus on our main aims and results. See specific responses below.

Spatial heterogeneity is not clearly defined and shown on the map (Fig.1). Importance of micro-topography is only stated towards the end of the discussion, but not mentioned earlier in the paper. Showing elevations in this large and complex wetland would help to define “zonation” announced in the title, and make presentation and interpretation of the data much clearer. Vegetation units, like “wet” and “mesic”, or four undefined “clusters” are not explained. Why two of these approaches are required? How do they add to each other? Would it be possible to combine all data from all surveys into new analysis based on the plots elevations? If positions of plots along the profiles can be accurately located, they can be easily linked to the topographic elevation values. Such allocation along the elevation gradient would link vegetation data to the physical properties of the site and justify “zonation” idea both in analysis and interpretation of the results. Currently wetness of the site is measured by the river levels from the distant gauge boards. Those values are the same for entire site. Elevation is only a proxy which can be applied to individual plots along all four profiles (data are freely available from LiDAR system) and reflect variation in wetness across the site.

Response: These comments were extremely helpful! We have now retrieved elevation data for each plot with LiDAR data and used that to illustrate the spatial heterogeneity in the new Fig. 1. We could then use elevation as a predictor (together with weather variables) in a more elaborate GLMM analysis of variation in Fritillaria flowering over time. This makes use of all six Fritillaria plots, replacing our previous simple comparison of plots 4 and 5. Furthermore, we could relate vegetation to elevation by superimposing elevation on the ordination (Fig. 5a)

 Rationale for the two approaches: In 1940 there was only Profile 1, so we wanted to analyse that profile separately to illustrate changes (in that part of the meadow) from 1940 to 2016. Data for Profiles 2-4 exist from 1982, so we could analyse the whole meadow (Profiles 1-4 from 1982 to 2016.

Management, which is undoubtedly one of the major drivers of vegetation composition, presented in this paper in a very descriptive way, which is difficult to follow. It needs to be formalised and linked clearly to both time line of the data collection and to the spatial heterogeneity of the site. Management is notoriously difficult to quantify and put into the data analysis. However, it is still possible as categorical data. If analysis is not possible, a role of management should be only mentioned in Introduction and in Discussion, in general terms. It can’t be outlined in the Conclusion as effects of management on vegetation were not studied in this paper.

Response: Unfortunately, the management history is poorly documented, and we agree that the description was complicated. We have now summarized the history in a table in Appendix S1. We were not able to perform any formal analysis of effects of management, but we now highlight some vegetation changes that could be explained by certain episodes in the management history: specifically the effects of too early grazing and of treatment with rotor cultivator in the wet parts. A main conclusion (also stressed by Reviewer 1) is that despite the changes in management, the vegetation seems rather resilient to change.

The second large aspect of this study is population analysis of Fritillaria meleagris, as one of the iconic species of Kungsängen. It would be most interesting to see the link between

the flowering dynamics of Fritillaria meleagris and vegetation changes studied in the same observation plots. If such analysis could’ve been done, the paper looked more uniform and linked. Again, population data analysed against elevation gradient would answer the question about effect of microtopography on population dynamics.

Response: Relevant point! The new GLMM analysis explicitly links Fritillaria dynamics to elevation. We don't have vegetation data for the Fritillaria plots, but we have now restructured the presentation of the vegetation along Profile 1. Previously the rank-abundance graph summarised the whole Profile, but now we separate the Wet and Mesic parts. This made it easier to discuss the link between Fritillaria and vegetation dynamics. For example, the peak in flowering shifted from the wetter plot 5 to the drier plot 4 between 1940 and 1982, and we now note (line 580) that this is 'consistent with the concurrent increased wetness indicated by vegetation changes in the wetter part in Profile 1 (related to the increased water level in the river between 1940 and 1970...).

Following the title of the paper, data analysis of the Fritillary population should be focused on spatial heterogeneity of the site. Differences in elevations between 6 observation plots should be quantified. Why and how only two out of six plots have been selected for a detailed analysis of effect of “wetness” ought to be clearly explained in Methods. Conclusion about Fritillary “wet” and “mesic” population’s loci reacting differently to the variation of the soil moisture in different seasons sounds very intriguing even correlations shown in Fig.9 are not strong. This observation deserves more focused discussion, based on authors’ knowledge of seasonal species growth.

Response: Thanks to the great suggestion to include elevation data for the plots we now made GLMM analyses with all six plots as suggested. We did this with both temperature and precipitation as predictors (together with elevation), and for the three periods of the year when Fritillaria is active. The results are quite informative: wet and cold periods favoured flowering in the drier part of the meadow, whereas dry and warm periods promoted flowering in the wetter parts. This could then explain that the peak in flowering shifted from the wetter plot 5 to the drier plot 4 between 1940 and 1982 (see previous point).

Referencing in this paper is very dense and intense. References occur even in Results where the findings of the current paper are presented. Very many quotes include the page number or the Figure number from other publications. That implies, the reader of the current paper has to keep at least a dozen of other publications opened, in order to follow the logic. References are not formatted according to Plos One requirement.

Response: We retained the page or figure number references only for the book about Kungsängen (Sandberg 1948 and Sernander 1948). Without these references it will be very difficult for a reader to trace and check the information. So it is only to facilitate for the reader, and can be removed if the editor so wishes. The ms is now formatted according to Plos One requirements.

The paper can be published after a major revision and re-writing. English language should be checked by native English speakers. In many places the writing is too descriptive. Numerous self-references to the previous most interesting and valuable publications of some authors, often overload the paper with details which are not directly relevant to the topic of this paper.

Response: The text is now checked by a native English speaking professional proofreader, and we have tried to simplify the text and removed details. 

The title: “Zonation” is a too general, not very explanatory term. It always require an adjective saying which factor has been “zoned”. Hydrology? Soil compaction? Soil mechanical composition? Grazing?

What is a difference/link between “zonation” and “spatial heterogeneity”?

There is no description/definition of the zones in the paper.

Response: We agree with this comment and now use only 'spatial heterogeneity' in title and most other places. We have replaced 'zonation' with 'wetness gradient' or 'elevation gradient' for a directional change where applicable. 

Line 22: “Vegetation clusters” is not a well known term, needs clarification concerning a principle point of clustering.

Response: Our view is that 'vegetation cluster' is a well known term. But we agree that a more detailed description of the methods used was needed. In Methods we have now added information about the functions we used in the widely used 'vegan' package in R. See response to the reviewer's comment on vegetation clustering below.

Line 25: “affected by weather”. It needs to be more specific about the timing of weather. Long-term? Previous year? Spring of the same year?

Response: We now clarify as "affected by temperature and precipitation in its active periods"

Line 50: broad-leaved grasses… What does it mean?

Response: Deleted.

Line 52: litter is a part of biomass. Directly, neither of them affects seed germination. Thick litter cut off an access to the ground, that should be explained.

Response: We don't agree that litter is part of biomass. According to Calow P (Ed.) 1998. The encyclopedia of ecology and environmental management (Blackwell Science) biomass is 'the total mass of living material' whereas litter would be the same as necromass. But we agree that the mechanisms should be explained so we now write (line 44): 'By shading and preventing seeds from reaching the soil surface this increase in biomass and litter makes it more difficult for plants to establish (Grime, 2001).'

Line 57: flooded habitats are not an “exception from the rules”. Hay cut still remove nutrients there. The sentence needs a different wording.

Response: We have rephrased as 'Removal of hay reduces nutrient levels, but in alluvial meadows productivity could be maintained as flooding brings new nutrients.'

Line 66: if Glyceria maxima is exotic to Swedish flora, that must be stated. Otherwise, this is a common aquatic species. Was an appearance of the species in the nature reserve caused by particular management?

Response: Deleted, not really relevant. 

Line 79: Vice versa – is not clear in the given context.

Response: Agreed, sentence simplified to 'For example, low management intensity might be disadvantageous for rosette-forming and early-flowering species'.

Line 101: Is “lily” a Swedish short name for Fritillary? If yes, that should be mentioned in the line 99. English name is snake’s head fritillary.

Response: Changed to F. meleagris.

Lines 123-124: the statement about perennial plants is not clear, needs re-writing. Why the allocation of resources in previous year reduces flowering?

Response: We referred to this as 'cost of reproduction' which is a well known concept, but we have tried to clarify as 'In perennial plants, the allocation of resources to reproduction may diminish the chances to flower next year. Such a cost of reproduction is difficult to demonstrate in natural populations, as it is often confounded by year-to-year variation in resources and weather conditions (Sletvold & Ågren, 2015)'.

Line 130: probability of individual flowering was estimated by Tatarenko (2019). Useful paper about Fritillary populations.

Response: We added reference to this paper. 

Lines 157-159: Is meadow situated on the floodplain? When it was at the sea level, was there floods from the sea? Not sure how the lake 6 km downstream influences the water table in the meadow. Were there any hydrological observations/measurements ever done?

Response: These details were not essential for the study and are deleted. The details of the hydrological history are presented in Appendix S1.

Line 183: “permanent annual mowing regime”? Either permanent or annual… Regular?

Response: Changed to 'mown annually'.

Lines 150-208: Location of the site should be described in Introduction, not in the Methods. Description of hydrology and management of the site is presented as a literature review, not as “Methods”. The information is very descriptive and difficult to follow to a reader who is not familiar with the site. If the data from the quoted publications are used in the vegetation and population analyses in this paper, they should be presented in the Table, separately for hydrology and especially, for management. Elements of Literature review should be moved to Introduction, where the site is described in details.

Response: As suggested the site description has been moved to Introduction. To facilitate reading we shortened the text, and added a table in Appendix S1 which should give a better overview.

The map of the site (Fig. 1) needs showing “zonation”, both hydrological and management-wise.

Response: The new Fig. 1 shows spatial variation in elevation. Legend to Fig. 1 now explains which area was affected by rotor cultivator in the 1990s.

The same applies to Vegetation “Methods”: having a Table showing which plots in which profile were surveyed in which year and by which botanist would allow much better understanding of the structure of botanical data.

Response: We have added such a table.

Vegetation clusters – a non-conventional approach to vegetation classification. It must be described in details in Methods. What are the criteria for definition of the clusters?

Response: As indicated in comment to Line 22 above we argue that clustering is not a non-conventional approach. But we have added details in methods: 

'Vegetation data were analysed using the ‘vegan’ ....'

Why two approaches of grouping the species in vegetation units were needed? One is “wet” and “mesic”, another – four clusters? Why a traditional vegetation classification was not used?

Response: We now more clearly describe that we grouped the plots in the first survey to define the starting point, i.e., the Wet and Mesic groups and from 1982 the four clusters. We then followed what later happened to the groups and the clusters. We did this to be able to follow changes in species composition in different parts of the meadow. 

 The wet-mesic grouping was made for Profile 1 in 1940. With only 23 plots along one profile we consider that a simple split according to elevation would suffice. As we now also have elevation data for all plots to support the approach we can see that the two groups are well defined and separated according to elevation (Mesic group, ≥ 1.55 m a.s.l.; Wet group, ≤ 1.35 m a.s.l.). The grouping is also supported by the NMDS in Fig. 2)

 The clustering was made for all profiles with the starting point 1982. We argue that this is a good description of the spatial variation over the whole meadow in 1982, and also in this case we support the approach with elevation data and write (line 322): 'We identified four clusters that were (i) spatially separated in the meadow (Fig. 1), (ii) well separated in the ordination described below (Fig. 5) and (iii) reflect the elevation gradient (mean elevation for Clusters 1-4 was 2.0, 1.9, 1.5 and 1.3 m a.s.l.).'

 It is possible that a traditional classification (meaning a phytosociological classification?) would have worked too, but we feel that our approaches defined the spatial variation well enough.

Lines 261-279 – again, this is not the Methods, this is a literature review on the species biology. This should be placed in Introduction. Some details (e.g., pollination, seasonal growth) are not relevant to the topic discussed in the paper.

Response: Moved to Introduction. As suggested, some details are deleted. However, the growth periods are relevant for the selection of periods in which weather could affect flowering, so parts of that are still required.

Lines 288-289 – the reference is a repeat of the same above in Line 163.

Response: The first instance is deleted. 

Line 307: give values for “highest”

Response: We replaced this with their overall mean cover (since these species did not change significantly between surveys. This section about changes in the whole profiles is toned down as the focus is now on changes within different parts of the meadow rather than in whole profiles. 

Line 312-313: Such prolonged gaps between observations don’t support conclusions about “directional” changes in this extremely dynamic vegetation.

Response: A valid comment, so we deleted the statements about 'directional' changes.

Line 318: Changes in Vegetation. Would it be a better title? “Development”, as well as “succession” would mean one-directional changes to a higher organisational level.

Response: We agree and changed heading to 'Changes in vegetation'

Line 331: What is a factor which drives communities “downwards”? The title of the paper is “zonation and spatial heterogeneity…” How does the spatial heterogeneity apply to the results described here?

Response: These changes are difficult to interpret in relation to spatial heterogeneity, and we now discuss them in relation to changes in grazing/mowing, disturbance and nitrogen deposition. 

Lines 349-380: changes in four clusters are presented in a very descriptive way, which is difficult to follow. Table 1 gives values of increase / decrease in species cover recorded in four surveys. As the main focus of the paper is the “spatial heterogeneity” (which is represented by “clusters” to large extent, isn’t it?), it would be more informative to group changes in species abundance according to the “clusters”.

Response: A valuable comment. Here we have made a total make-over! The previous version had analyses and rank-abundance diagrams that covered the whole data set. In agreement with the comment we now present changes separately for the clusters in the rank-abundance graphs. Thereby, focus on heterogeneity becomes stronger. The new graphs also meant that we could shorten the text presentation of the clusters, hopefully making it easier to follow. We retain the previous Table 1 as supplementary information (Appendix S7) for the floristically inclined reader.

Line 383-388: levels of significance should be shown.

Response: We added to the text that analyses with p-values are in Appendices S3 and S4. 

Line 383-384: “higher” than what?

Response: Changed to 'higher for plots in the Wet group than in the Mesic group'

Line 421: “precipitation” is mentioned here for the first time. All weather factors should be mentioned in Methods. Was precipitation actually measured/analysed?

Response: In the previous version we only used precipitation in a preliminary analysis (which was poorly explained) and instead used water flow in the river as the wetness proxy. However, we do not have water flow data for 1938, so during the revision we concluded that it was a shame that we could not include the 1938 flower counts in the models, so we have now clarified that we use temperature and precipitation data from the same source. 

Lines 433-465: this information belongs to Introduction, no discussion of the findings of the current paper here.

Response: Now mentioned in the Introduction only.

Line 466: Variations in management against changes in species richness should be first presented in the Results section. That would justify discussion presented here.

Response: As the Reviewer noted above, management is difficult to quantify and analyse. We now present a rather general hypothesis (line 139): 'the management has periodically adopted the modern practice of early hay harvest, as well as being mechanised, and we hypothesise that this (in combination with other environmental changes, such as nitrogen deposition) may have led to losses in species richness and plant diversity and to changes in community composition.' We link back to this in the opening of the Discussion (line 434): 'We had expected a decrease in species richness and diversity as the management has varied, with increased mechanisation, periods with too early mowing and grazing, and occasional tussock reduction.'

Line 479: “zonation” was not mentioned in Results

Response: We deleted the word 'zonation'.

Lines 529-534: very difficult to follow logic here

Response: We have tried to clarify as (line 491): 'Under a cutting regimen C. acuta will not form tussocks (Almquist, 1929). Without cutting and with periodic high water, tussocks are formed that provide micro-habitats for mesic meadow species (Kołos & Banaszuk, 2018), explaining why from 1995 a few meadow species are found in the wettest part of the meadow.'

Line 536: “root cultivation” needs to be explained

Response: 'Rotor cultivator' seems to be the term used for the machine and write: 'The wettest parts were only mown in some years, allowing the sedges to form tall tussocks that had to be cut by rotor cultivator in the 1990s'.

Line 546-547: the question about “dynamics of population or”… is not clear.

Response: Sentence deleted.

Line 548-557: how does it link to the topic of the paper?

Response: See next point.

Line 555: purpose for re-analysis of Zang (1983) dataset should be explained in Introduction. As supplementary material explains, only dormancy has been added to the published analysis. Why that was required? No new data were collected recently. Assumption of plants being dead after 2 years of not being seen above the ground is not justified. Two-year dormancy in Liliaceae and Orchidaceae is as common and one-year dormancy. This analysis doesn’t add any new interpretation of the plant distribution/dynamics along the hydrological gradient.

If a use of this analysis had been justified in Introduction, model should be explained in Methods, or reference to the Supplementary material given there. Data from the analysis should be shown in the Results, of reference to Supplementary material given there. After that it can be discussed here.

Response: In methods we now clarify that we need the matrix analysis (particularly the sensitivity analysis) to be able to discuss the importance of the different life-stages (particularly flowering) for the population. We also clarify that Zhang did not perform a matrix analysis (but published the data that we now use for such an analysis). The relevant findings are now described in Results, with reference to Supplement for details. An interpretation of how the dormant stage may influence future population dynamics is now included. It would have been most interesting to repeat the data and do the analyses along the hydrological gradient (as wisely hinted by the referee) but the dataset is too small for this.

Line 563-564: Not quite correct extraction of information from Tatarenko et al 2022. Fig 6 there doesn’t show “flowering and vegetative adults”. It shows ratio between flowering and vegetative plants, not “vegetative adults”.

Response: Well spotted, thanks! We now give comparable numbers for the different studies (percent flowering of the total population).

Line 558-571: How does it relate to the topic of the paper?

Response: In the aims we now more clearly introduce the question of the importance of flowering for the population dynamics. Since variation in the number of flowers among years is a topic of the paper we think that it is relevant to discuss that this variation is caused more by variation in the propensity to flower than in variation in total population size. 

Line 578: How exactly do fritillary plants “track long-term changes in wetness”?

Response: This phrase was unclear and we changed to "track variation in wetness". Since we did not study the physiological mechanisms we cannot answer how this happens "exactly". But in essence the GLMM analyses show that wet and cold periods favoured flowering in the drier part of the meadow, and vice versa. And in the long-term, the peak in flowering shifted from the wetter plot 5 to the drier plot 4 between 1940 and 1982 (Fig. 8), consistent with the concurrent increased wetness in vegetation in the wetter part.

Line 604-616: mainly repeat of old data and a small and non-conclusive finding from the current project discussed against the data for not taxonomically close species. This doesn’t add much to the main story discussed in the paper.

Response: We introduced 'cost of reproduction' in introduction and aims as a potential reason for variation in flowering propensity among years, so we discuss this negative finding. Since 'cost of reproduction' is an important concept in the literature that discusses the probability for a plant to flower it seems odd not to include it in our discussion. We have tried to clarify this both in Introduction and here in Discussion. Even if not closely related, we find the comparisons between Fritillaria meleagris and the two orchid species interesting as they highlight the difficulty in detecting cost of reproduction. Even if they are not taxonomically close, they share some features: similar habitat, long-lived, irregular flowering. 

Line 617-628: not relevant to the topic of the current paper.

Response: We now clarify why this is interesting in the Introduction, which was easier now as we followed the suggestion to move up the description of Fritillaria biology from Methods to Introduction(see comment to Line 261). We also repeat the reason why it is interesting in the Discussion, to make it clearer to the reader.

---

## [Decision Letter · Decision Letter 1]

5 Jan 2023

PONE-D-22-16087R1Spatial heterogeneity ensures long-term stability in vegetation and Fritillaria meleagris flowering in Uppsala Kungsäng, a semi-natural meadowPLOS ONE

Dear Dr. Rydin,

Thank you for submitting your manuscript to PLOS ONE. After careful consideration, we feel that it has merit but does not fully meet PLOS ONE’s publication criteria as it currently stands. Therefore, we invite you to submit a revised version of the manuscript that addresses the points raised during the review process.

Both reviewers note the revised manuscript is significantly improved over the original submission and acknowledge your efforts to address all their concerns.  Most remaining issues deal with problems of clarity and flow/structure, although I identify a couple other needed areas for improvement in the list below.  Abstract.  Structure, content, and clarity are lacking.  There are many guides and publications available that one can consult to get a good sense of the format of a strong Abstract, e.g. https://guides.lib.uci.edu/c.php?g=334338&p=2249902.  Also, some text is strangely worded, which adds to the lack of clarity, e.g. Lines 18-21.  What exactly do you mean by active periods?  Do you mean when the species is flowering?  Regardless, doesn't temperature and precipitation dictate active periods/flowering versus the other way around?  As written, the content of the sentence seems backwards.  In the next sentence, what do you mean by 'Thereby'?  I assume what you're trying to say is that because of this annual and spatial variation in flowering due to temp, precip, and wet/dry areas of the meadow, there was significant year to year variation in flowering.  But maybe not.  Be explicit rather than using a short hand word like 'thereby'.  Line 23-26.  Add a comma after 'composition' and 'the' before 'long-term'.  That will help clarify that richness and composition refer to the meadow vegetation and not the *Fritillaria* population (e.g. Reviewer 1 comment that populations don't have richness).  You might also consider rewording to make it even more explicit that the first part of that sentence refers to meadow vegetation.  Also, seems 'long-term in...' should be 'long-term of...'  What do you mean by 'upheld'?Materials & Methods.  Clarity/flow of this section would be improved by additional structure in the form of subsections.  As Reviewer 2 noted, statistical approaches are commonly included in a separate subsection.Species names.  On first use and when used at the beginning of a sentence fully spell out both Genus and species.  On subsequent use, abbreviate Genus, e.g. *Carex acuta* when first stated, *C. acuta* elsewhere unless it starts a sentence.  Please check that your use of all species names follows this rule.White morph.  For clarity be explicit that 'flower-picking townsfolk' are agents of natural selection on flower color because of their preference for white flowers (e.g. Reviewer 1 comment).Line 252-253.  Why is font size different here?References.  Italicize species names throughout (they currently aren't).  Line 831.  Delete extra space before 'term'.  Ensure that spacing and format used is consistent throughout, e.g. page numbering isn't, Line 838 vs 840.

We look forward to receiving your revised manuscript.

Kind regards,

Dr. Janice L. Bossart

Academic Editor

PLOS ONE

Journal Requirements:

Reviewers' comments:

Reviewer's Responses to Questions

**Comments to the Author**

1. If the authors have adequately addressed your comments raised in a previous round of review and you feel that this manuscript is now acceptable for publication, you may indicate that here to bypass the “Comments to the Author” section, enter your conflict of interest statement in the “Confidential to Editor” section, and submit your "Accept" recommendation.

Reviewer #1: All comments have been addressed

Reviewer #2: All comments have been addressed

2. Is the manuscript technically sound, and do the data support the conclusions?

Reviewer #1: Yes

Reviewer #2: Yes

3. Has the statistical analysis been performed appropriately and rigorously? 

Reviewer #1: Yes

Reviewer #2: I Don't Know

4. Have the authors made all data underlying the findings in their manuscript fully available?

Reviewer #1: Yes

Reviewer #2: Yes

5. Is the manuscript presented in an intelligible fashion and written in standard English?

Reviewer #1: Yes

Reviewer #2: Yes

6. Review Comments to the Author

Reviewer #1: This manuscript is much improved from the previous draft and suggested edits have been addressed. The addition of elevation from lidar is very valuable and the discussion is more interesting and broader. The manuscript is still quite long and detailed but rejigging the methods would help improve the flow. I have some minor comments below, which are largely for clarity and improving understanding for the readers.

Abstract

I think the abstract is lacking a so what, why part. Your first sentence is a general one about management but then the next is about analysing the vegetation. There needs to be a sentence in the middle bringing together what your study is addressing.

Line 14: This wording is slightly odd, perhaps something like “Management is crucial for maintaining the richness and diversity of semi-natural grasslands”.

Line 33: I don’t think you need to say “overall”

Introduction

Line 45: I wouldn’t mention “such as the one studied in this paper”. Its nice to keep the first paragraph in the introduction general.

Line 46: You mention several elements but only state two – mowing and grazing.

Line 52: Do you have a reference to support this?

Line 55: What is a general impoverishment of flora? Loss of species diversity? Loss of species richness?

Line 63: Reference needed here. You could also add here that meadows and semi-natural grasslands are important for the delivery of ecosystems e.g. pollination, carbon storage etc. See Bengtsson et al (2019) Ecosphere.

Figure 1 is much improved. However, I would show the elevation scale with a legend on the map rather than written in the legend, as its quite harder for a reader to understand what the elevation of yellow areas are for example.

Line 101-103: These sentences seem a bit random. Perhaps is worth moving these until after you’ve described the history of the site.

Method

I think it would be clearer to rejig the methods. Personally, I think it would be clearer to have “1. Vegetation Survey” – here include the general vegetation survey and F.meleagris survey too. Then have “2. Statistical analysis” – this could be split in 2.1 Vegetation change and 2.2. Variation in flowering.

Line 313: This would be more appropriately placed in the statistical analysis, when you mention about using elevation.

Line 346: What source did you use for the six indicators? Reference needed.

Line 401: Good to provide a reference for why you centred and scale predictor variables e.g. Schielzeth, H. 2010. Simple means to improve the interpretability of regression coefficients. – Methods Ecol. Evol. 1: 103–113.

394: So for climate were both temperature and precipitation used? Did you check if these were correlated?

Line 406: Perhaps make this a bit clearer by stating we eliminated these three years and thus the analysis was based on the remaining 12 years of data.

Results

Table 1: I’m assuming this is the average species richness and diversity? It would be good to say that and provide standard error or deviation with this.

Fig 2: You’ve labelled the plots now so you can remove the “w” and “m” part from your written figure legend. It seems a bit random to just “include some other species mentioned in the text”, I would be tempted to just leave it as the 6 most abundant (same for Figure 5a).

Fig 9: Why were these elevation values selected?

Fig 10: How were high and low elevation split?

Discussion

This section is much improved.

Line 714: Vegetation change or change in vegetation would be more consistent with your results subheadings.

Line 750-762: It might be nitrogen deposition and grazing/mowing are interacting. Its hard to disentangle the two – see Wilson E, Wells T, Sparks T (1995) Are calcareous grasslands in the UK under threat from nitrogen deposition? -an experimental determination of a critical load. J Ecol 83:823–832

Line 1062: And I guess the importance of ensuring that some management takes place even if its not the traditional management that used to take place?

Reviewer #2: Good work on revision of the paper!

I still feel it contains at least two papers in one. Still too much emphasis on management, in places. Still, no biological justification to include flower color analysis in the paper. Ratio of white/pink flowers is population dependent, genetically driven, nothing to do with picking up white flowers by visitors of the meadow.

An opening sentence of the Abstract states importance of management, while the title of the paper: role of spatial heterogeneity. It would be logical to start Abstract with the statement relevant to the topic.

"Species richness, overall species composition and long-term stability in the Fritillaria meleagris population"

----- A population does not have species richness or species composition…

Introduction has been largely improved, looks much better. Only, once again, management there has been stated very strongly from the very beginning, while hydrology and spatial heterogeneity are not presented as the main drivers of the vegetation in the meadows. I still would suggest to focus more on the topic. E.g., floodplain meadows developed substantial heterogeneity within their landscape… ancient channels, flood water ponds, natural elevation gradient, etc.

They have been used as prime habitats for hay making in summer and grazing in… , sustainable biomass production because of nutrients arriving with the floods,… Lack of hay cut/grazing – deposit of dead biomass (litter) preventing seedling recruitments…

Line 597-598 and below

Again, a very interesting discussion about weather effect on flowering of Fritillaria meleagris. The species seasonal development is very well studied quite long time ago. See

http://www.fritillariaicones.com › info › Baranova

If to discuss weather effects on population dynamics, it would be most useful to apply existing knowledge about different life stages to weather at different times of year to follow actual relationship between two. I think, this is a good data and idea for paper on its own.

7. PLOS authors have the option to publish the peer review history of their article (what does this mean?). If published, this will include your full peer review and any attached files.

Reviewer #1: No

Reviewer #2: No

---

## [Author Response · Author response to Decision Letter 1]

26 Jan 2023

Dear Editor,

We now submit a second revised version of the manuscript PONE-D-22-16087, entitled 'Spatial heterogeneity ensures long-term stability in vegetation and Fritillaria meleagris flowering in Uppsala Kungsäng, a semi-natural meadow'. Below we describe how we have dealt with each of the comments from the Editor and the two reviewers. 

For the author team,

Håkan Rydin

Comments from the Editor

Thank you for submitting your manuscript to PLOS ONE. After careful consideration, we feel that it has merit but does not fully meet PLOS ONE’s publication criteria as it currently stands. Therefore, we invite you to submit a revised version of the manuscript that addresses the points raised during the review process.

Both reviewers note the revised manuscript is significantly improved over the original submission and acknowledge your efforts to address all their concerns. Most remaining issues deal with problems of clarity and flow/structure, although I identify a couple other needed areas for improvement in the list below. 

• Abstract. Structure, content, and clarity are lacking. There are many guides and publications available that one can consult to get a good sense of the format of a strong Abstract, e.g. https://guides.lib.uci.edu/c.php?g=334338&p=2249902. 

Response: Thanks for the hint! The Abstract is re-written with Editor’s and Reviewers’ comments in mind.

Also, some text is strangely worded, which adds to the lack of clarity, e.g. Lines 18-21. What exactly do you mean by active periods? Do you mean when the species is flowering? Regardless, doesn't temperature and precipitation dictate active periods/flowering versus the other way around? As written, the content of the sentence seems backwards. In the next sentence, what do you mean by 'Thereby'? I assume what you're trying to say is that because of this annual and spatial variation in flowering due to temp, precip, and wet/dry areas of the meadow, there was significant year to year variation in flowering. But maybe not. Be explicit rather than using a short hand word like 'thereby'. 

Response: We have replaced ‘active periods’ with defining the phenological phases of the plant, so it should now be clear that weather in these periods (June previous year, September previous year, and March to April) affects flowering (in May). The sentence with ‘thereby’ is re-phrased.

Line 23-26. Add a comma after 'composition' and 'the' before 'long-term'. That will help clarify that richness and composition refer to the meadow vegetation and not the Fritillaria population (e.g. Reviewer 1 comment that populations don't have richness). You might also consider rewording to make it even more explicit that the first part of that sentence refers to meadow vegetation. Also, seems 'long-term in...' should be 'long-term of...' What do you mean by 'upheld'? 

Response: Changed to “Species richness and species composition of the meadow vegetation, and the long-term stability of the F. meleagris population are maintained by…”.

• Materials & Methods. Clarity/flow of this section would be improved by additional structure in the form of subsections. As Reviewer 2 noted, statistical approaches are commonly included in a separate subsection. 

Response: We have re-arranged Material and methods as suggested by Reviewer 2

• Species names. On first use and when used at the beginning of a sentence fully spell out both Genus and species. On subsequent use, abbreviate Genus, e.g. Carex acuta when first stated, C. acuta elsewhere unless it starts a sentence. Please check that your use of all species names follows this rule. 

Response: Fixed.

• White morph. For clarity be explicit that 'flower-picking townsfolk' are agents of natural selection on flower color because of their preference for white flowers (e.g. Reviewer 1 comment). 

Response: It was actually Reviewer 2 who commented on this. We clarify that the flower-picking “may have decreased its relative frequency in the population by reducing recruitment from seeds”. 

• Line 252-253. Why is font size different here? 

Response: Fixed.

• References. Italicize species names throughout (they currently aren't). Line 831. Delete extra space before 'term'. Ensure that spacing and format used is consistent throughout, e.g. page numbering isn't, Line 838 vs 840.

Response: Fixed.

Reviewers' comments:

1. If the authors have adequately addressed your comments raised in a previous round of review and you feel that this manuscript is now acceptable for publication, you may indicate that here to bypass the “Comments to the Author” section, enter your conflict of interest statement in the “Confidential to Editor” section, and submit your "Accept" recommendation.

Reviewer #1: All comments have been addressed

Reviewer #2: All comments have been addressed

 2. Is the manuscript technically sound, and do the data support the conclusions?

Reviewer #1: Yes

Reviewer #2: Yes

 3. Has the statistical analysis been performed appropriately and rigorously? 

Reviewer #1: Yes

Reviewer #2: I Don't Know

4. Have the authors made all data underlying the findings in their manuscript fully available?

Reviewer #1: Yes

Reviewer #2: Yes

5. Is the manuscript presented in an intelligible fashion and written in standard English?

Reviewer #1: Yes

Reviewer #2: Yes

Reviewer #1: 

This manuscript is much improved from the previous draft and suggested edits have been addressed. The addition of elevation from lidar is very valuable and the discussion is more interesting and broader. The manuscript is still quite long and detailed but rejigging the methods would help improve the flow. I have some minor comments below, which are largely for clarity and improving understanding for the readers.

Abstract

I think the abstract is lacking a so what, why part. Your first sentence is a general one about management but then the next is about analysing the vegetation. There needs to be a sentence in the middle bringing together what your study is addressing.

Response: The Abstract is re-written with Editor’s and Reviewers’ comments in mind. We have now toned down the management aspect in the first sentence to more directly focus on the research question: that the vegetation may have been affected by environmental change and altered management. 

Line 14: This wording is slightly odd, perhaps something like “Management is crucial for maintaining the richness and diversity of semi-natural grasslands”.

Response: Deleted as a result of the previous comment.

Line 33: I don’t think you need to say “overall”

Response: Deleted.

Introduction

Line 45: I wouldn’t mention “such as the one studied in this paper”. Its nice to keep the first paragraph in the introduction general.

Response: Deleted.

Line 46: You mention several elements but only state two – mowing and grazing.

Response: Simplified as “traditional management usually involves mowing in mid- to late summer and aftermath grazing” 

Line 52: Do you have a reference to support this?

Response: We have added a reference.

Line 55: What is a general impoverishment of flora? Loss of species diversity? Loss of species richness?

Response: Changed to “loss of species diversity and a more trivial flora”.

Line 63: Reference needed here. You could also add here that meadows and semi-natural grasslands are important for the delivery of ecosystems e.g. pollination, carbon storage etc. See Bengtsson et al (2019) Ecosphere.

Response: Thanks, we incorporated ecosystem services, supported by this reference.

Figure 1 is much improved. However, I would show the elevation scale with a legend on the map rather than written in the legend, as its quite harder for a reader to understand what the elevation of yellow areas are for example.

Response: Elevation scale added.

Line 101-103: These sentences seem a bit random. Perhaps is worth moving these until after you’ve described the history of the site.

Response: Yes, we did so.

Method

I think it would be clearer to rejig the methods. Personally, I think it would be clearer to have “1. Vegetation Survey” – here include the general vegetation survey and F.meleagris survey too. Then have “2. Statistical analysis” – this could be split in 2.1 Vegetation change and 2.2. Variation in flowering.

Response: Re-arranged as suggested.

Line 313: This would be more appropriately placed in the statistical analysis, when you mention about using elevation.

Response: Done

Line 346: What source did you use for the six indicators? Reference needed.

Response: Reference to Tyler’s paper added.

Line 401: Good to provide a reference for why you centred and scale predictor variables e.g. Schielzeth, H. 2010. Simple means to improve the interpretability of regression coefficients. – Methods Ecol. Evol. 1: 103–113.

Response: Nice reference, since the answer is in its title! We added it.

394: So for climate were both temperature and precipitation used? Did you check if these were correlated?

Response: We have now added that the two predictors were uncorrelated (with p-values)

Line 406: Perhaps make this a bit clearer by stating we eliminated these three years and thus the analysis was based on the remaining 12 years of data.

Response: Changed as suggested.

Results

Table 1: I’m assuming this is the average species richness and diversity? It would be good to say that and provide standard error or deviation with this.

Response: No, since this table is meant to show overall changes it is total species richness in these plots. So we changed the caption to “Total species richness…”.

Fig 2: You’ve labelled the plots now so you can remove the “w” and “m” part from your written figure legend. It seems a bit random to just “include some other species mentioned in the text”, I would be tempted to just leave it as the 6 most abundant (same for Figure 5a).

Response: labels “w” and “m” are removed. The choice of species may seem random, but they are not arbitrarily selected. In the text we mention, for example, ruderal species (and were asked by reviewer to add some more), and it is crucial for the interpretation to show these in the ordination. To avoid the appearance of randomness, we changed to “some other indicative species (such as ruderals)”. Same in Fig. 5.

Fig 9: Why were these elevation values selected?

Response: Not selected; the numbers (1.83, 1.84 etc.) are the actual elevation values for each plot. We think this is clear from the legend.

Fig 10: How were high and low elevation split?

Response: We did not 'split' the dataset into high and low elevation plots. Rather, elevation was treated as a continuous variable in our models. To better visualize the interactive effects of climate and elevation, we show predictions of our model across the climate axis for the highest and lowest elevations included in the data.

Discussion

This section is much improved.

Line 714: Vegetation change or change in vegetation would be more consistent with your results subheadings.

Response: Changed to “Changes in vegetation” as in Methods and Results.

Line 750-762: It might be nitrogen deposition and grazing/mowing are interacting. Its hard to disentangle the two – see Wilson E, Wells T, Sparks T (1995) Are calcareous grasslands in the UK under threat from nitrogen deposition? -an experimental determination of a critical load. J Ecol 83:823–832

Response: True, so we added “Nitrogen effects are often confounded with effects of management and can be difficult to disentangle” with the suggested reference (and one more).

Line 1062: And I guess the importance of ensuring that some management takes place even if its not the traditional management that used to take place?

Response: The last sentence is changed to “…highlighting the importance of spatial heterogeneity and appropriate management…”.

 

Reviewer #2: 

Good work on revision of the paper! I still feel it contains at least two papers in one. Still too much emphasis on management, in places. 

Response: We toned down management in the opening of the abstract. See also response to comment on the Introduction below. 

Still, no biological justification to include flower color analysis in the paper. Ratio of white/pink flowers is population dependent, genetically driven, nothing to do with picking up white flowers by visitors of the meadow.

Response: We agree with the Editor’s argument that selective ‘flower-picking townsfolk' are agents of natural selection on flower color because of their preference for white flowers. In the text we clarify that this selective picking of white flowers “may have decreased its relative frequency in the population by reducing recruitment from seeds”.

An opening sentence of the Abstract states importance of management, while the title of the paper: role of spatial heterogeneity. It would be logical to start Abstract with the statement relevant to the topic.

Response: The Abstract is re-written with Editor’s and Reviewers’ comments in mind. We have now toned down the management aspect in the first sentence to more directly focus on the research question: that their vegetation may have been affected by environmental change and altered management.

"Species richness, overall species composition and long-term stability in the Fritillaria meleagris population" ----- A population does not have species richness or species composition…

Response: Changed to “Species richness and species composition of the meadow vegetation, and the long-term stability of the F. meleagris population…”. 

Introduction has been largely improved, looks much better. Only, once again, management there has been stated very strongly from the very beginning, while hydrology and spatial heterogeneity are not presented as the main drivers of the vegetation in the meadows. I still would suggest to focus more on the topic. E.g., floodplain meadows developed substantial heterogeneity within their landscape… ancient channels, flood water ponds, natural elevation gradient, etc. They have been used as prime habitats for hay making in summer and grazing in… , sustainable biomass production because of nutrients arriving with the floods,… Lack of hay cut/grazing – deposit of dead biomass (litter) preventing seedling recruitments…

Response: We have made an attempt to restructure the Introduction along these lines. 

Line 597-598 and below

Again, a very interesting discussion about weather effect on flowering of Fritillaria meleagris. The species seasonal development is very well studied quite long time ago. See

http://www.fritillariaicones.com › info › Baranova

If to discuss weather effects on population dynamics, it would be most useful to apply existing knowledge about different life stages to weather at different times of year to follow actual relationship between two. I think, this is a good data and idea for paper on its own.

Response: We have added the suggested reference in the Introduction. We agree that another paper is possible, and we are considering to model flowering in response to future climate scenarios.

---

## [Editor Report · Decision Letter 2]

3 Feb 2023

PONE-D-22-16087R2

Spatial heterogeneity ensures long-term stability in vegetation and Fritillaria meleagris flowering in Uppsala Kungsäng, a semi-natural meadow

PLOS ONE

Dear Dr. Rydin,

Thank you for submitting your manuscript to PLOS ONE.  Although revisions have greatly improved the manuscript, there are a few very minor issues that remain to be addressed before it can be moved forward.  Please revise accordingly and resubmit.

Captions for Figure 2 & 5:  "...some other indicative species..."  Like Reviewer 1, I stumble over the phrasing and meaning here.  For one, please change "indicative' to 'indicator'.  Indicator species is a commonly used phrase but indicative species is not.  Second,  the use of "...some other..." doesn't fully clarify the problem that Reviewer 1 identified.  I tried toying around with some modifications, but wasn't completely happy with any.  Maybe: "...six most abundant species in each envelope and other representative indicator species that were mentioned in the text (such as ruderals)".  See if you can further tweak text for Caption Fig 2 and 5 to increase clarity.  Regardless, the same text should be added to both caption, e.g. Fig 5 caption doesn't currently include the clarifying bit of text, "such as ruderals".   

Line 305:  R.acris.  Insert a space after the period, delete the space after 's'.

Line 316:  R.auricomus.  Insert a space after the period.

Line 647:  Please change 'upheld' to 'maintained'.

Please also carefully read through a clean version of your revised manuscript before resubmitting.  Closely check for any errors and that all references have been correctly cited within the text and references cited section.  PLOS ONE does not use a copy editor.

We look forward to receiving your revised manuscript.

Kind regards,

Dr. Janice L. Bossart

Academic Editor

PLOS ONE
---

## [Author Response · Author response to Decision Letter 2]

7 Feb 2023

Dear Editor,

We now submit a third revised version of the manuscript PONE-D-22-16087, entitled 'Spatial heterogeneity ensures long-term stability in vegetation and Fritillaria meleagris flowering in Uppsala Kungsäng, a semi-natural meadow'. Below we describe how we have dealt with each of the comments from the Editor. 

For the author team,

Håkan Rydin

Comments from the Editor

Thank you for submitting your manuscript to PLOS ONE. Although revisions have greatly improved the manuscript, there are a few very minor issues that remain to be addressed before it can be moved forward. Please revise accordingly and resubmit.

Captions for Figure 2 & 5: "...some other indicative species..." Like Reviewer 1, I stumble over the phrasing and meaning here. For one, please change "indicative' to 'indicator'. Indicator species is a commonly used phrase but indicative species is not. Second, the use of "...some other..." doesn't fully clarify the problem that Reviewer 1 identified. I tried toying around with some modifications, but wasn't completely happy with any. Maybe: "...six most abundant species in each envelope and other representative indicator species that were mentioned in the text (such as ruderals)". See if you can further tweak text for Caption Fig 2 and 5 to increase clarity. Regardless, the same text should be added to both caption, e.g. Fig 5 caption doesn't currently include the clarifying bit of text, "such as ruderals". 

Response: We appreciate your efforts in finding a way to clarify what we mean. We follow your suggestion and have changed to “…and other representative species that are mentioned in the text (such as ruderals).” We avoid using “indicator” as this word is associated with Ellenberg indicator values in the manuscript; these species are in the text.

Line 305: R.acris. Insert a space after the period, delete the space after 's'.

Response: Corrected.

Line 316: R.auricomus. Insert a space after the period.

Response: Corrected.

Line 647: Please change 'upheld' to 'maintained'.

Response: Changed as suggested.

---

## [Editor Report · Decision Letter 3]

8 Feb 2023

Spatial heterogeneity ensures long-term stability in vegetation and Fritillaria meleagris flowering in Uppsala Kungsäng, a semi-natural meadow

PONE-D-22-16087R3

Dear Dr. Rydin,

We’re pleased to inform you that your manuscript has been judged scientifically suitable for publication and will be formally accepted for publication once it meets all outstanding technical requirements.

Kind regards,

Dr. Janice L. Bossart

Academic Editor

PLOS ONE
---

## [Editor Report · Acceptance letter]

27 Feb 2023

PONE-D-22-16087R3 

Spatial heterogeneity ensures long-term stability in vegetation and *Fritillaria meleagris* flowering in Uppsala Kungsäng, a semi-natural meadow 

Dear Dr. Rydin:

I'm pleased to inform you that your manuscript has been deemed suitable for publication in PLOS ONE. Congratulations! Your manuscript is now with our production department. 

Kind regards, 

on behalf of

Dr. Janice L. Bossart 

Academic Editor

PLOS ONE